# Genome-wide CRISPRi screening identifies OCIAD1 as a prohibitin client and regulatory determinant of mitochondrial Complex III assembly in human cells

Maxence Le Vasseur[1], Jonathan Friedman[1,2], Marco Jost[3,4,5†], Jiawei Xu[1], Justin Yamada[1], Martin Kampmann[3,4,6,7], Max A Horlbeck[3,4], Michelle R Salemi[8], Brett S Phinney[8], Jonathan S Weissman[3,4,9,10], Jodi Nunnari[1]*

[1]Department of Molecular and Cellular Biology, College of Biological Sciences, University of California, Davis, Davis, United States; [2]Department of Cell Biology, University of Texas Southwestern Medical Center, Dallas, United States; [3]Department of Cellular and Molecular Pharmacology, University of California at San Francisco, San Francisco, United States; [4]Howard Hughes Medical Institute, University of California at San Francisco, San Francisco, United States; [5]Department of Microbiology and Immunology, University of California at San Francisco, San Francisco, United States; [6]Institute for Neurodegenerative Diseases and Department of Biochemistry and Biophysics, University of California at San Francisco, San Francisco, United States; [7]Chan-Zuckerberg Biohub, San Francisco, United States; [8]Proteomics Core Facility, University of California, Davis, Davis, United States; [9]Whitehead Institute, Cambridge, United States; [10]Department of Biology, Massachusetts Institute of Technology, Cambridge, United States

*For correspondence:
jmnunnari@ucdavis.edu

Present address: †Department of Microbiology, Harvard Medical School, Boston, United States

**Abstract** Dysfunction of the mitochondrial electron transport chain (mETC) is a major cause of human mitochondrial diseases. To identify determinants of mETC function, we screened a genome-wide human CRISPRi library under oxidative metabolic conditions with selective inhibition of mitochondrial Complex III and identified ovarian carcinoma immunoreactive antigen (OCIA) domain-containing protein 1 (OCIAD1) as a Complex III assembly factor. We find that OCIAD1 is an inner mitochondrial membrane protein that forms a complex with supramolecular prohibitin assemblies. Our data indicate that OCIAD1 is required for maintenance of normal steady-state levels of Complex III and the proteolytic processing of the catalytic subunit cytochrome $c_1$ (CYC1). In OCIAD1 depleted mitochondria, unprocessed CYC1 is hemylated and incorporated into Complex III. We propose that OCIAD1 acts as an adaptor within prohibitin assemblies to stabilize and/or chaperone CYC1 and to facilitate its proteolytic processing by the IMMP2L protease.

## Introduction

Mitochondria are double membrane-bound organelles of endosymbiotic origin that produce most of the ATP in eukaryotic cells through oxidative phosphorylation (OXPHOS) (*Mitchell, 2011*). OXPHOS depends on the mitochondrial electron transport chain (mETC), which transfers electrons from NADH and succinate to molecular oxygen. The mETC is comprised of a series of four large inner mitochondrial membrane (IMM) complexes (CI–CIV) that assemble into supercomplexes of defined

stoichiometry (*Letts and Sazanov, 2017*). Substrate oxidation-driven electron transfer is coupled to the translocation of protons across the IMM to generate an electrochemical gradient harvested by the ATP synthase (CV) for ATP production. In addition, the mETC and the associated tricarboxylic acid (TCA) cycle support a network of metabolic functions. The mETC helps maintain the redox balance of carrier pairs involved in hundreds of biochemical reactions (*Luna-Sánchez et al., 2017*; *Titov et al., 2016*; *Wang and Hekimi, 2016*; *Ying, 2008*; *Ziosi et al., 2017*), a basic requisite for sustaining metabolism in living cells, and is also essential for generating the proton gradient that drives the import of nuclear-encoded mitochondrial proteins across the IMM (*Eilers et al., 1987*; *Martin et al., 1991*; *Pfanner and Neupert, 1986*; *Schleyer et al., 1982*). Perturbing the assembly or function of the mETC can lead to multisystem mitochondrial disorders (*Chinnery, 1993*; *Rodenburg, 2016*; *Tucker et al., 2013*; *Wanschers et al., 2014*) and is linked to more general pathologies, such as diabetes (*Antoun et al., 2015*; *Ramírez-Camacho et al., 2020*), neurodegeneration (*Devi et al., 2008*; *Giachin et al., 2016*; *Keeney et al., 2006*), heart diseases (*Andreu et al., 2000*; *Casademont and Miró, 2002*; *Hagen et al., 2013*; *Valnot et al., 1999*), and cancer (*Hoekstra and Bayley, 2013*; *Janeway et al., 2011*; *Pantaleo et al., 2014*; *Urra et al., 2017*; *Van Vranken et al., 2015*).

The biogenesis of the mETC requires the concerted expression of nuclear and mitochondrial DNA (mtDNA) encoded genes and is highly regulated. Coordination of mETC subunits of dual origin occurs in part via the formation of modular intermediates within mitochondria that assemble sequentially into functional complexes (*Aich et al., 2018*; *Guerrero-Castillo et al., 2017*; *Lobo-Jarne et al., 2020*; *Ndi et al., 2018*; *Stephan and Ott, 2020*; *Van Vranken et al., 2015*). Assembly of mETCs strategically occurs in specialized domains that link protein import, membrane insertion, and assembly machineries (*Singh et al., 2020*; *Stoldt et al., 2018*). Prohibitins are thought to promote mETC assembly and quality control by assembling into inner membrane ring-like scaffold structures that specify local protein and lipid composition (*Nijtmans et al., 2000*; *Singh et al., 2020*). In mammalian cells, prohibitins associate with a variety of inner membrane proteins, including mitochondrial translocases, subunits of mETC, the DnaJ-like chaperone DNACJ19, and matrix-AAA (m-AAA) proteases (*Nijtmans et al., 2000*; *Richter-Dennerlein et al., 2014*; *Steglich et al., 1999*; *Yoshinaka et al., 2019*). The interaction of prohibitin with these key assembly and quality control proteins either directly modulates their activities and/or influences their client interactions to regulate and potentially coordinate a plethora of mitochondrial functions.

Here, we use an unbiased genome-wide CRISPRi approach to screen for human genes modulating the cellular response to antimycin A, a chemical inhibitor of mitochondrial Complex III. Complex III, also called ubiquinol-cytochrome $c$ oxidoreductase or cytochrome $bc1$, is centrally situated within the mETC. Complex III is an obligate homodimeric enzyme ($CIII_2$) embedded in the inner membrane with each monomer composed of 10–11 subunits. Only three subunits contain catalytically active redox groups: cytochrome b (MT-CYB), cytochrome $c_1$ (CYC1), and the Rieske iron–sulfur protein (UQCRFS1), with other accessory subunits that likely stabilize the assembly (*Lee et al., 2001*; *Malaney et al., 1997*). We identified ovarian carcinoma immunoreactive antigen domain-containing protein 1 (OCIAD1), a poorly characterized protein, as a key regulator of Complex III biogenesis. Our data indicate that OCIAD1 is a client of prohibitin supramolecular assemblies and is required for the IMMP2L-dependent proteolytic processing of the catalytic subunit CYC1. Thus, we postulate that within prohibitin assemblies, OCIAD1 facilitates CYC1 proteolytic processing by the IMMP2L.

## Results

### Genome-wide CRISPRi screen for antimycin sensitivity identifies Complex III molecular determinants

CRISPR screens have emerged as a powerful approach to identify key genes regulating molecular processes in human cells (*Gilbert et al., 2014*; *Jost et al., 2017*; *To et al., 2019*). To identify regulatory determinants of mitochondrial function, we screened for genes that either sensitized or protected against antimycin A, a selective inhibitor of mitochondrial respiratory Complex III. Candidate genes were identified using a genome-scale CRISPRi screen performed in human K562 cells stably expressing the dCas9-KRAB transcriptional repressor (*Gilbert et al., 2013*). Cells were infected with the hCRISPRi-v2 sgRNA pooled library containing 10 sgRNAs per gene (*Horlbeck et al., 2016*) and

grown for 6 days in glucose-free media containing galactose, which favors oxidative metabolism over glycolysis. The cell population was then halved and subjected to four cycles of treatment with either vehicle or antimycin A (3.5–3.75 nM; 24 hr treatment, 48 hr post-washout recovery), which created a growth difference of ~3–4 doublings between treated and untreated cells (*Figure 1A*). Following the final recovery phase, cells were harvested at ~750 cells per sgRNA and sgRNA-encoding cassettes were PCR-amplified from genomic DNA. The abundance of each individual sgRNA was then quantified by next-generation sequencing and a phenotype score (ρ) was calculated for each

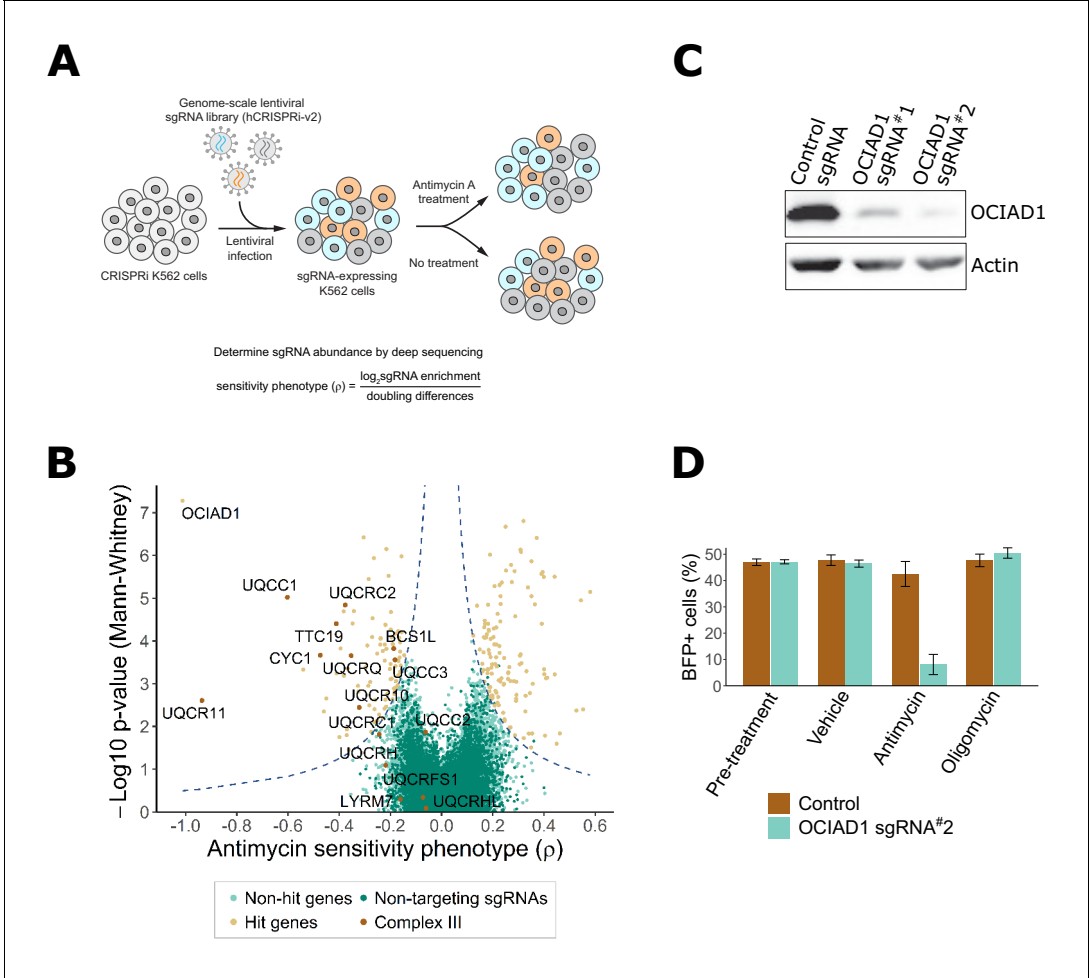

**Figure 1.** Genome-scale CRISPRi antimycin screen identifies genes regulating mitochondrial Complex III. (**A**) Schematic overview of the genome-wide CRISPRi screen. K562 dCas9 cells stably expressing dCas9-KRAB were infected with a pooled genome-scale sgRNA library. After growth in galactose, cells were subjected to four pulses of antimycin A or vehicle treatment followed by a 48 hr recovery period. After the last antimycin A pulse, genomic DNA from each condition was isolated and sgRNA abundance was quantified by deep sequencing. (**B**) Volcano plot showing the statistical significance (y axis) versus phenotype scores (ρ, x axis) of control non-targeting and genome-wide targeting sgRNAs. Knockdown of Complex III structural proteins and assembly factors sensitized cells to antimycin A. Genes were considered a hit if they scored above a threshold of ρ z-score x—log₁₀ p-value of 7 (dashed line). (**C**) CRISPRi knockdown of ovarian carcinoma immunoreactive antigen domain-containing protein 1 (OCIAD1) expression. Western blot showing the expression level of OCIAD1 in K562 dCas9-KRAB cells stably expressing either a control non-targeting sgRNA or two different sgRNAs against OCIAD1. CRISPRi-based silencing reduced OCIAD1 protein expression by ~90%. (**D**) Validation of the OCIAD1 phenotype. K562 dCas9 cells were mixed with an equal number of K562 dCas9-KRAB BFP + cells stably expressing a non-targeting sgRNA (brown bars) or an sgRNA against OCIAD1 (light blue bars). Cell mixtures were then treated with the drug or a vehicle for 24 hr. The percentage of BFP + cells in the cell mixtures was measured by flow cytometry before and 24 hr after treatment. OCIAD1 silencing selectively sensitized cells to antimycin treatment.

The online version of this article includes the following figure supplement(s) for figure 1:

**Figure supplement 1.** Silencing genes related to Complex I, pyruvate, and tricarboxylic acid (TCA) metabolism protect cells against chemical inhibition of Complex III.

**Figure supplement 2.** Silencing Complex III genes aggravate the cellular response to antimycin A.

gene as described (*Gilbert et al., 2014*; *Jost et al., 2017*; *Kampmann et al., 2013*). This phenotype score represents the differential pressure each sgRNA exerts on cell growth in the presence versus absence of antimycin A. Positive ρ values indicate protection and negative ρ values indicate sensitization to antimycin A.

Using this approach, we identified 217 genes that significantly modulated sensitivity to antimycin A under oxidative conditions (*Figure 1B*, *Supplementary file 1*). Knockdown of 128 of these genes protected against antimycin A. Gene ontology (GO) enrichment analysis performed on this group identified an enrichment of genes encoding for mitochondrial respiratory chain Complex I. Complex I is the most upstream entry point into the electron transport chain and is composed of 44 unique subunits, 37 of which are encoded by the nuclear DNA with the remaining seven subunits encoded by the mitochondrial genome (*Fiedorczuk et al., 2016*; *Guerrero-Castillo et al., 2017*). Knockdown of about one-sixth of the nuclear-encoded Complex I subunits, as well as additional assembly factors, significantly protected against antimycin A treatment (*Figure 1—figure supplement 1A–C*). Complex I subunit hits were distributed on all Complex I assembly modules except the proximal portion of the peripheral arm, indicating that the protective response is likely dependent on a general loss of Complex I function. Knockdown of genes encoding components of the TCA cycle also protected against antimycin A treatment, including those encoding enzymes that participate in both forward flux through this pathway to maintain OXPHOS and reverse flux for reductive carboxylation. Other protective hits included a protective gene encoding an assembly factor of Complex II, which connects the TCA cycle to the respiratory chain, upstream of Complex III, as well as genes encoding the mitochondrial pyruvate carrier and pyruvate dehydrogenase, which connects glycolysis with the TCA cycle (*Figure 1—figure supplement 1D,E*). It was recently reported that the loss of mitochondrial Complex I activity suppressed toxicity caused by oligomycin, an ATP synthase inhibitor, and to a lesser extent by antimycin A, by promoting glycolysis and reductive carboxylation (*To et al., 2019*). However, the suppressive effect we observe is potentially inconsistent with this mechanism as our screen was performed under different metabolic conditions that promote oxidative metabolism and suppress glycolysis. Thus, it is possible that the mechanism of antimycin A toxicity suppression in our screen was a consequence of a reduction in respiratory chain activity upstream of Complex III to protect against production of ROS, further suggesting that multiple suppressive mechanisms for antimycin toxicity may exist, dependent on cellular metabolic status.

In our screen, knockdown of 89 genes sensitized cells to antimycin A treatment (*Figure 1B*), including 9 of the 15 nuclear-encoded Complex III subunits or assembly factors (*Figure 1—figure supplement 2*). Consistent with this, GO enrichment analysis identified Complex III as the most enriched term for antimycin A toxicity (*Figure 1—figure supplement 2A*). These data validate the screen and confirm that the mechanism of growth inhibition by antimycin A was a consequence of Complex III inhibition.

In addition to genes encoding Complex III, OCIAD1 was identified as a strongly sensitizing hit (*Figure 1B*). OCIAD1 encodes a poorly characterized predicted transmembrane protein (Figure 3A) that is aberrantly expressed in ovarian carcinomas and implicated in the regulation of mitochondrial metabolism via Complex I (*Shetty et al., 2018*). We validated the antimycin A-sensitizing phenotype of OCIAD1 by performing a growth competition assay in K562 cells using CRISPRi cell lines stably expressing an individual sgRNA against OCIAD1 (sgRNA#2). This sgRNA was identified in our screen and effectively silenced OCIAD1 expression (*Figure 1C*). Silencing OCIAD1 selectively compromised growth of antimycin A-treated cells, but not growth of oligomycin-treated cells (*Figure 1D*), suggesting, together with our screen data, that OCIAD1 knockdown specifically sensitizes cells to inhibition of Complex III.

## OCIAD1 is required for the assembly of Complex III

We assessed whether OCIAD1 regulates the assembly and/or stability of mitochondrial respiratory complexes using blue-native polyacrylamide gel electrophoresis (BN-PAGE), followed by Western blotting using antibodies directed against core constituents of respiratory Complexes I–V (*Figure 2*). Mitochondria were isolated from K562 control or OCIAD1 knockdown cells grown in galactose and respectively expressing a non-targeting sgRNA or sgRNA#2 against OCIAD1. We also analyzed mitochondria isolated from K562 OCIAD1 knockdown cells in which OCIAD1 expression had been reintroduced to near endogenous levels using lentiviral delivery (*Figure 2A*). There were no significant defects observed in the assembly of Complex I, II, IV, or V (*Figure 2B,C,E and F*). By contrast,

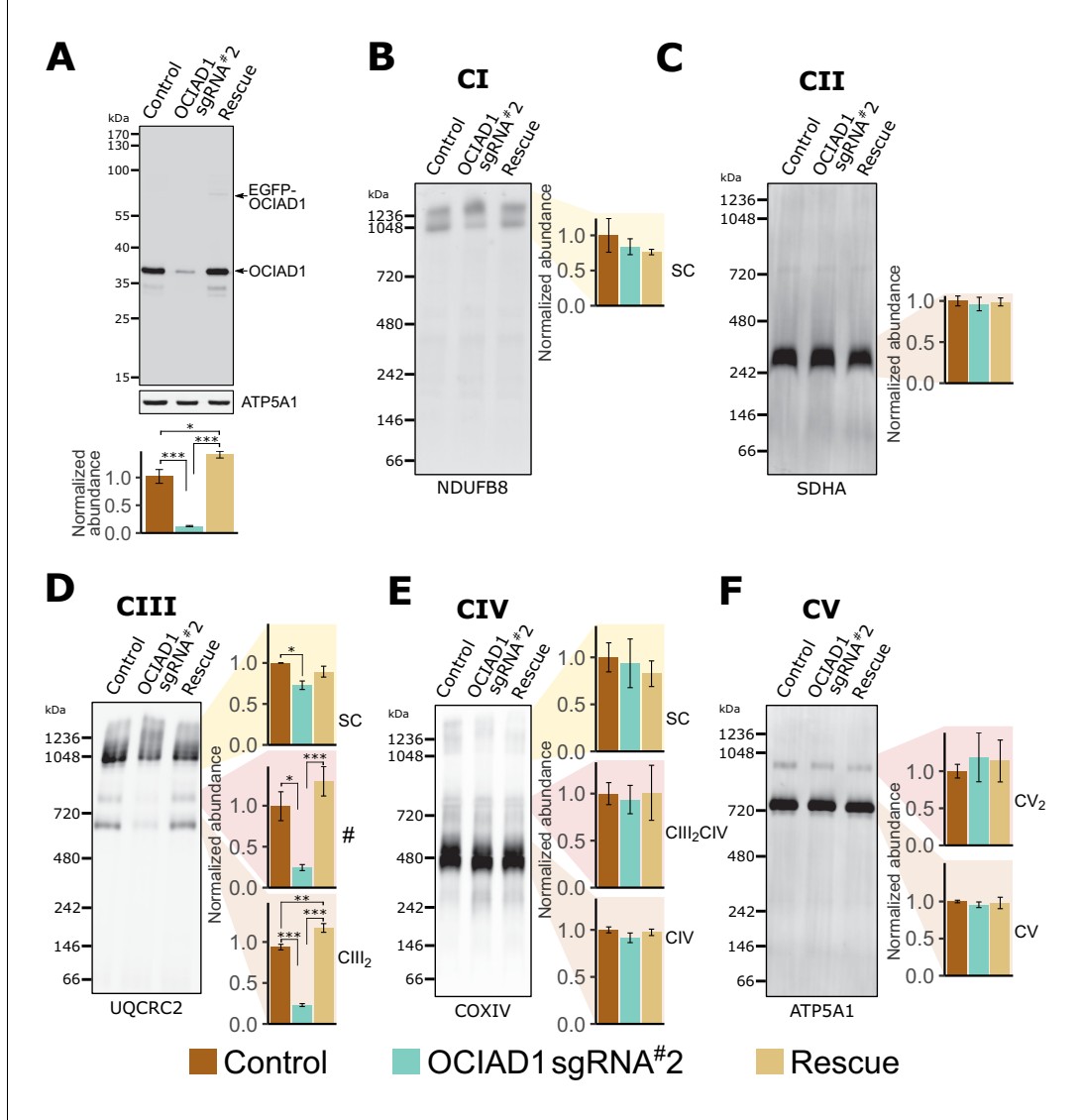

**Figure 2.** Ovarian carcinoma immunoreactive antigen domain-containing protein 1 (OCIAD1) is required for CIII$_2$ assembly. (**A**) Western blot showing CRISPRi silencing of OCIAD1 protein expression (12.47 ± 1.06% of control) in K562 cells. Rescue of OCIAD1 (141.20 ± 6.07% of control) by lentivirus transduction with a P2A multicistronic vector with high cleavage efficiency (98.89 ± 0.12%). The upper band (EGFP-OCIAD1) represents intact fusion gene product. ATP5A1 served as loading control. (**B–F**) OCIAD1 is selectively required for Complex III assembly. Blue-native polyacrylamide gel electrophoresis (BN-PAGE) analysis of digitonin-solubilized mitochondria followed by Western blotting using NDUFB8 (Complex I), SDHA (Complex II), UQCRC2 (Complex III), COXIV (Complex IV), and ATP5A1 (ATP synthase). The ATP5A1 signal from monomeric CV (**F**) was used as a loading control to quantify UQCRC2 intensities (**D**) as both proteins were probed on the same membrane. In panel B, both bands were used for quantification. Values represent normalized intensity ± SEM (n = 3 biological replicates). Asterisks (*p<0.05, **p<0.01, or ***p<0.001) correspond to the adjusted (false discovery rate [FDR]) p-values from the post-ANOVA (analysis of variance) pairwise t-test.

we observed a selective defect in Complex III assembly in cells depleted of OCIAD1 (*Figure 2D*). The abundance of Complex III was significantly reduced in mitochondria from OCIAD1 knockdown cells and restored to wildtype levels by OCIAD1 reintroduction, indicating that this defect is specific to loss of OCIAD1 function (*Figure 2D*, CIII$_2$). At steady state, Complex III is an obligate dimer (CIII$_2$) that participates with Complexes I and IV (CI and CIV) to form higher-order assemblies in mitochondria, known as supercomplexes (CIII$_2$CIV, CICIII$_2$, CICIII$_2$CIV). We also observed a smaller but significant reduction in CIII$_2$ supercomplex assemblies (*Figure 2D*, SC). Mitochondrial respiratory chain complexes proteins are unusually long-lived (*Fornasiero et al., 2018*) and thus, the smaller impact of OCIAD1 on supercomplexes might be a consequence of enhanced stability of supercomplexes

compared to individual complexes. In mitochondria depleted of OCIAD1, we also observed a reduction in a species whose mass/migration was consistent with the $CIII_2CIV$ supercomplex (*Figure 2D*, #). However, we did not detect a coincidental decrease in abundance of the co-migrating Complex IV species by Western blot analysis of COX4, a Complex IV marker (compare # and $CIII_2CIV$ in *Figure 2D and E*). The identity of this higher-order OCIAD1-sensitive Complex III species remains unknown.

## OCIAD1 is a mitochondrial inner membrane protein

OCIAD1 is annotated as a mitochondrial protein by the MitoCarta 3.0 inventory (*Rath et al., 2021*), but also has been reported to localize to endosomes and peroxisomes (*Antonicka et al., 2020*; *Mukhopadhyay et al., 2003*; *Sinha et al., 2018*). Consistent with the MitoCarta repository, we found that OCIAD1 primarily localized to mitochondria in U2OS cells, as evidenced by indirect

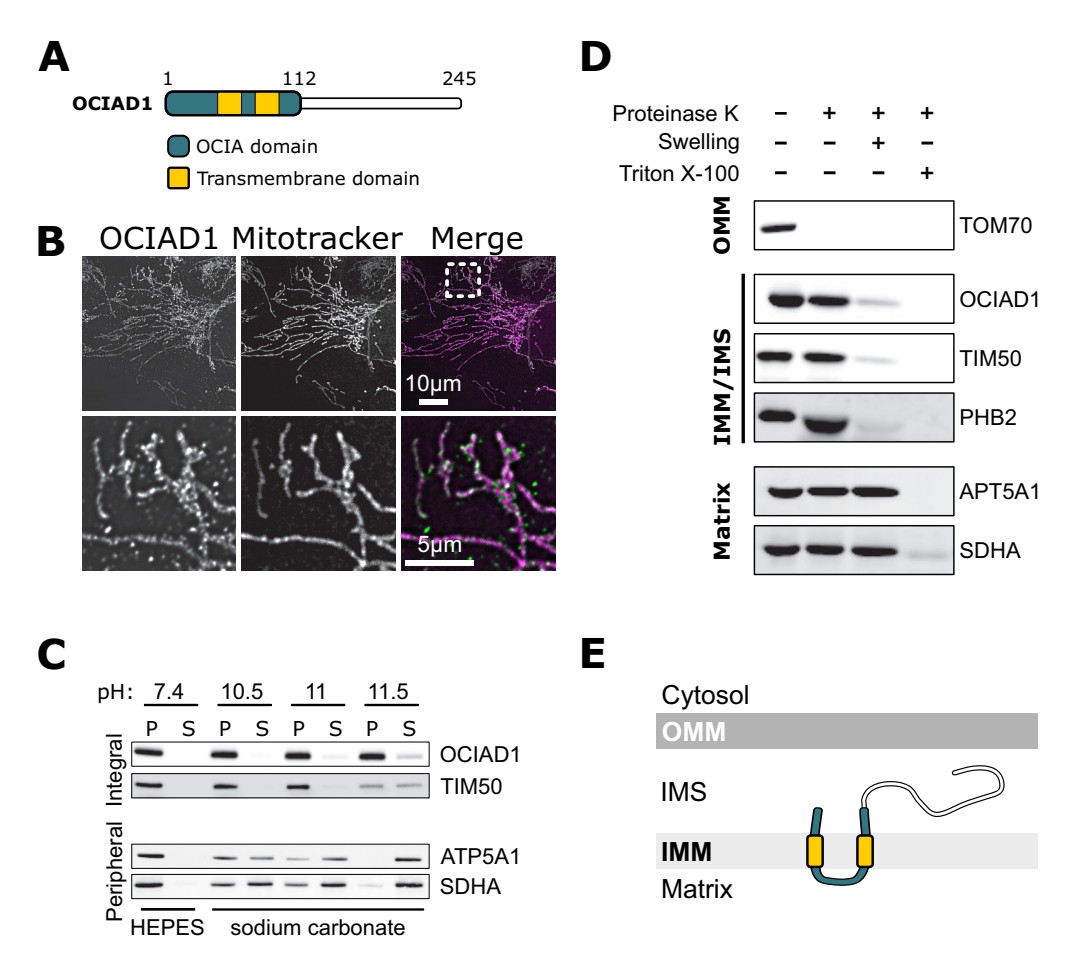

**Figure 3.** Ovarian carcinoma immunoreactive antigen domain-containing protein 1 (OCIAD1) is an inner mitochondrial membrane protein. (**A**) Schematic illustration of OCIAD1 domain organization. (**B**) Representative images of fixed U2OS cells stained with Mitotracker (magenta) and immunolabeled using anti-OCIAD1 antibodies (green). Lower panel is a magnification of the inset shown in the upper panel. (**C**) OCIAD1 is an integral membrane protein. Sodium carbonate extraction fractions (pH 10.5–11.5) immunoblotted with anti-OCIAD1, anti-TIM50, anti-ATP5A1, and anti-SDHA antibodies. P and S indicate pellet and soluble fractions, respectively. (**D**) OCIAD1 localizes to the inner membrane. Protease protection assay fractions immunoblotted with anti-OCIAD1, anti-prohibitin 2 (PHB2), anti-TIM50, anti-ATP5A1, and anti-SDHA antibodies. (OMM: outer mitochondrial membrane, IMM: inner mitochondrial membrane, IMS: intermembrane space). (**E**) Schematic illustration of OCIAD1 topology within the inner membrane.

The online version of this article includes the following figure supplement(s) for figure 3:

**Figure supplement 1.** Ovarian carcinoma immunoreactive antigen domain-containing protein 1 (OCIAD1) termini are localized in the mitochondrial intermembrane space.

immunofluorescence analysis using validated polyclonal OCIAD1 antibodies (*Figure 3B*). Following extraction of peripheral membrane proteins with carbonate treatment of increasing pH, OCIAD1 and the known inner membrane protein TIM50 (*Yamamoto et al., 2002*) both remained in the membrane pellet fraction isolated by differential centrifugation (*Figure 3C*). In contrast, the peripheral membrane proteins ATP5A1 and SDHA were readily extracted and found in the supernatant (*Figure 3C*). These data indicate that OCIAD1 is an integral membrane protein, consistent with the presence of two predicted transmembrane domains (*Figure 3A*). Recently, OCIAD1 was suggested to reside in the outer mitochondrial membrane (OMM) based on proximity labeling (*Antonicka et al., 2020*; *Lee et al., 2017*). To further investigate the localization of OCIAD1, we conducted proteinase K protection assays on freshly isolated mitochondria. Whereas the validated OMM protein TOM70 was digested by treatment of intact mitochondria with proteinase K, OCIAD1 was resistant to degradation (*Figure 3D*). OCIAD1 was however susceptible to proteolytic degradation after compromising OMM integrity by hypo-osmotic treatment, similar to other IMM proteins (*Figure 3D*). Overall, these results demonstrate that OCIAD1 is an integral IMM protein.

Next, we sought to determine the topology of the OCIAD1 protein within the IMM using proteinase K as the amino acid sequence between the two predicted OCIAD1 transmembrane domains contains predicted proteinase K cleavage sites. Analysis of OCIAD1 deletion constructs by Western blotting analysis using polyclonal rabbit anti-OCIAD1 antibodies identified the last 25 amino acids of the OCIAD1 C-terminus as the antigenic determinant (*Figure 3—figure supplement 1A,B*). OCIAD1 proteolytic fragments were not observed by Western analysis of mitoplasts treated with proteinase K (*Figure 3—figure supplement 1C*). Thus, given the location of the OCIAD1 epitope, this observation suggests that the C-terminus was degraded and therefore localized to the intermembrane space (IMS). We also tested OCIAD1 localization and topology using a bipartite split GFP complementation assay, in which the 11 stranded β-barrel GFP fluorophore is reconstituted from a separately expressed N-terminal β-strands ($GFP_{1-10}$) and a C-terminal 16 amino acid β-strand ($GFP_{11}$), as previously described (*Hyun et al., 2015*). Specifically, we created U2OS cells stably expressing $GFP_{1-10}$ targeted either to the matrix or IMS and transiently expressed proteins tagged with $GFP_{11}$. We validated this system by expressing known matrix and IMS proteins, CoQ9 and MICU1, respectively, with C-terminal $GFP_{11}$ tags and measuring the efficiency of GFP complementation by flow cytometry (*Figure 3—figure supplement 1D*). CoQ9-$GFP_{11}$ complemented GFP only when expressed in matrix-targeted $GFP_{1-10}$ cells, consistent with its matrix localization (*Johnson et al., 2005*). Conversely, MICU1-$GFP_{11}$ only produced GFP signal when expressed in IMS-targeted $GFP_{1-10}$ cells, consistent with its localization to the IMS (*Hung et al., 2014*; *Tsai et al., 2016*). With a validated topology assay in hand, we transiently expressed OCIAD1 tagged with $GFP_{11}$ at either the N- or C-terminus and observed GFP signal only in the IMS-targeted $GFP_{1-10}$ cells (*Figure 3—figure supplement 1D*). Together, these data indicate that OCIAD1 is a transmembrane protein embedded in the mitochondrial inner membrane with its N- and C-termini facing the IMS (*Figure 3E*).

## OCIAD1 interacts with the supramolecular prohibitin complex

To gain insight into how OCIAD1 facilitates steady-state Complex III assembly, we mapped its interactome using affinity enrichment mass spectrometry (AE-MS). We immunopurified OCIAD1 from DSP-crosslinked cell lysates prepared from K562 OCIAD1 knockdown cells and K562 OCIAD1 knockdown cells rescued with wildtype OCIAD1 and analyzed the eluates by label-free quantitative mass spectrometry (*Figure 4A*, *Figure 4—source data 1*). We identified Complex III subunits and assembly factors, which supports our BN-PAGE data indicating that OCIAD1 regulates Complex III assembly (*Figure 4A*, in green). In addition, we identified subunits of the prohibitin complex, PHB1 and PHB2, as potential OCIAD1 interactors (*Figure 4A*, in dark purple), consistent with previously published work (*Richter-Dennerlein et al., 2014*). We also identified several prohibitin interactors, including C1QBP, COX4I1, and DNAJC19, the mitochondrial m-AAA proteases AFG3L2 and SPG7, as well as the AFG3L2-interactor MAIP1, and the protease IMMP2L, all previously identified in published studies examining the prohibitin interactome (*Richter-Dennerlein et al., 2014*; *Yoshinaka et al., 2019*; *Figure 4A*, in light green). Prohibitins form large hetero-oligomeric ring complexes composed of assemblies of PHB1/PHB2 dimers in the inner membrane of mitochondria (*Tatsuta et al., 2005*). These complexes are thought to constitute molecular scaffolds that define functional domains to regulate the lateral distribution of membrane lipids and proteins within the IMM (*Osman et al., 2009*; *Richter-Dennerlein et al., 2014*). Prohibitin structures associate with the

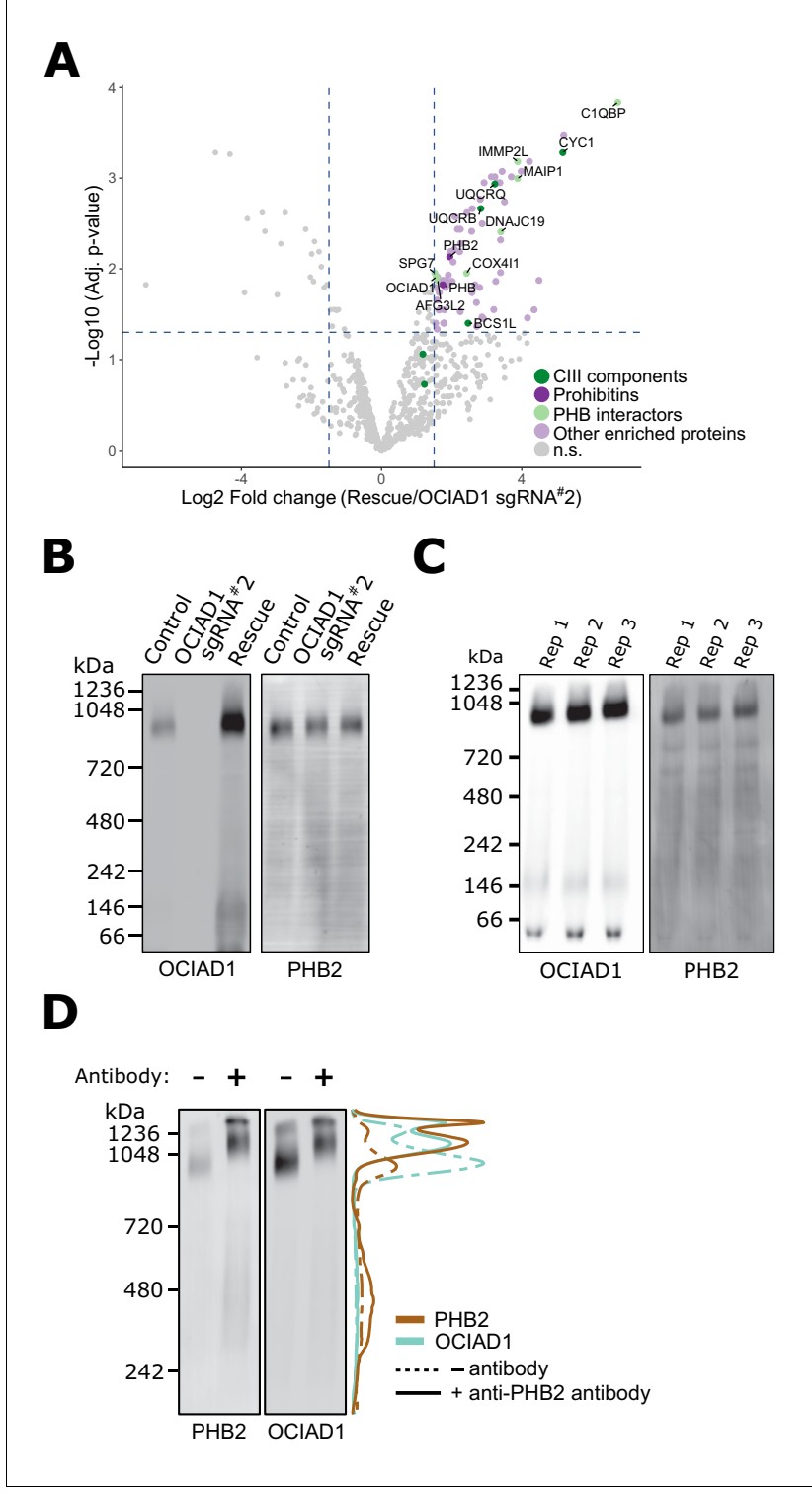

**Figure 4.** Ovarian carcinoma immunoreactive antigen domain-containing protein 1 (OCIAD1) forms a complex with prohibitin supramolecular assemblies. (**A**) Volcano plot showing the statistical significance ($-\log_{10}$ false discovery rate [FDR] adjusted p-value; y axis) versus $\log_2$ fold change (x axis) of proteins enriched in OCIAD1 pull-down performed on DSP-crosslinked K562 cell lysates from OCIAD1 knockdown cells and OCIAD1 knockdown cells rescued with wildtype OCIAD1. Proteins with a $\log_2$ fold change $\geq 1.5$ and an adjusted p-value $< 0.05$ were considered significantly enriched (n = 3 biological replicates, n.s. = non significantly enriched). (**B**) Blue-native polyacrylamide gel electrophoresis (BN-PAGE) of lauryl maltose neopentyl glycol (LMNG)

*Figure 4 continued on next page*

*Figure 4 continued*

detergent-solubilized mitochondrial membranes isolated from U2OS control, OCIAD1 knockdown, and OCIAD1 knockdown cells rescued with wildtype OCIAD1. The membrane was immunoblotted with anti-OCIAD1 and anti-PHB2 antibodies. (C) BN-PAGE of LMNG detergent-solubilized mitochondrial membranes isolated from U2OS control cells (n = 3 biological replicates) and immunoblotted with anti-OCIAD1 and anti-PHB2 antibodies. Electrophoresis was stopped before elution of the migration front to calculate the fraction of OCIAD1 that associates with PHB2 assemblies (66.91 ± 0.35%). (D) Mitochondria from K562 cells solubilized with LMNG and pre-incubated with anti-Phb2 antibodies (solid line) or vehicle (dotted line) were analyzed by BN-PAGE and immunoblotted with anti-OCIAD1 and anti-prohibitin two antibodies. Line scan traces represent the distribution profile of P (brown) and OCIAD1 (light blue).

The online version of this article includes the following source data and figure supplement(s) for figure 4:

**Source data 1.** Output of the limma statistical analysis of the OCIAD1 interactome in K562 OCIAD1 knockdown cells and K562 OCIAD1 knockdown cells rescued with wildtype OCIAD1.

**Source data 2.** Output of the limma statistical analysis of the whole proteome analysis by LC-MS/MS.

**Figure supplement 1.** The ovarian carcinoma immunoreactive antigen domain-containing protein 1 (OCIAD1) paralog, OCIAD2, localizes to the mitochondria inner membrane.

**Figure supplement 2.** Ovarian carcinoma immunoreactive antigen domain-containing protein 1 ( OCIAD1) and OCIAD2 paralogs are functionally divergent.

**Figure supplement 3.** The role of ovarian carcinoma immunoreactive antigen domain-containing protein 1 (OCIAD1) in CIII$_2$ assembly is independent of cell type and glucose availability.

**Figure supplement 4.** Ovarian carcinoma immunoreactive antigen domain-containing protein 1 (OCIAD1) regulates steady-state levels of Complex III subunits.

**Figure supplement 5.** The distal region of the ovarian carcinoma immunoreactive antigen (OCIA) domain is essential for the function of OCIA domain-containing protein 1 (OCIAD1) in CIII$_2$ assembly.

---

inner membrane m-AAA protease to modulate their activity in both the specific processing of inner membrane proteins and the targeted degradation of unassembled inner membrane proteins (*Bonn et al., 2011*; *Ehses et al., 2009*; *Koppen et al., 2009*; *Li et al., 2019*; *Merkwirth et al., 2008*; *Steglich et al., 1999*).

We also examined the molecular features of native OCIAD1 by BN-PAGE to assess potential interactors. Specifically, lauryl maltose neopentyl glycol (LMNG)-solubilized mitochondria isolated from K562 control and OCIAD1 knockdown cells, as well as K562 rescue cells expressing wildtype OCIAD1, were analyzed by BN-PAGE followed by Western analysis using anti-OCIAD1 antibodies (*Figure 4B*, left panel). This analysis demonstrated that the majority of OCIAD1 associates with large 0.9–1 MDa supramolecular assemblies that co-migrate with prohibitin (*Figure 4B,C*). To test whether OCIAD1 and prohibitin interact within the 0.9–1 MDa supramolecular assemblies, we performed an in-gel mobility shift assay. For this purpose, LMNG-solubilized mitochondrial membranes from K562 cells were incubated with either vehicle alone or with anti-PHB2 antibody and subjected to BN-PAGE, followed by Western analysis using anti-PHB2 or OCIAD1 antibodies. Pre-incubation of mitochondrial membranes with anti-PHB2 antibody, but not vehicle, retarded the migration of both PHB2 and OCIAD1 supramolecular assemblies to a similar degree (*Figure 4D*, compare dashed and solid lane line scans, respectively). Thus, together our data indicate that OCIAD1 associates with prohibitin complexes in the IMM.

## The OCIAD1 paralog, OCIAD2, has similar topology and interacts with prohibitin but is not functionally redundant with OCIAD1

OCIAD1 has a paralog in vertebrates, OCIAD2, which likely arose from tandem gene duplication of a common ancestor around 435–500 million years ago (*Sinha et al., 2018*). The paralogs share domain structure and significant homology (*Figure 4—figure supplement 1A*) and have been reported to hetero-oligomerize (*Sinha et al., 2018*), suggesting a shared function. Using indirect immunofluorescence, carbonate extraction, and protease protection analysis, we showed that, as expected, OCIAD2 localized to mitochondria with a topology similar to OCIAD1 (*Figure 4—figure supplement 1B–D*).

We examined whether OCIAD2, like OCIAD1, functions in Complex III assembly. OCIAD2 was not a hit in our antimycin A screen (*Figure 4—figure supplement 2A*) and Western blot analysis indicated that the K562 cells used in our CRISPRi screen do not express OCIAD2 (*Figure 4—figure*

supplement 2B). Therefore, we used U2OS cells, which express both paralogs, and generated individual and double OCIAD1/OCIAD2 knockdown cell lines, by identifying OCIAD2 shRNAs that efficiently silenced OCIAD2 expression (*Figure 4—figure supplement 2C*, shRNA1-3). Similar to our results in K562 cells, expression of sgRNA#2 against OCIAD1 effectively suppressed OCIAD1 expression in U2OS cells (0.86 ± 0.43% of control) but did not affect OCIAD2 expression (104.38 ± 18.48% of control; *Figure 4—figure supplement 2D*). Conversely, expressing an shRNA targeting OCIAD2 selectively silenced OCIAD2 expression in U2OS cells (8.17 ± 2.75% of control), but did not alter OCIAD1 levels (92.89 ± 8.46% of control; *Figure 4—figure supplement 2D*). We next examined the abundance of $CIII_2$ in the different cell lines by BN-PAGE analysis of mitochondria isolated from cells grown in glucose-free media containing galactose. Knockdown of OCIAD1 in U2OS cells decreased steady-state levels of $CIII_2$ relative to control cells, consistent with our observations in K562 cells. In contrast, knockdown of OCIAD2 did not affect $CIII_2$ levels (*Figure 4—figure supplement 2E*). Given that this analysis was performed on cells grown in glucose-free media containing galactose, we considered whether the role of OCIAD1 and OCIAD2 in $CIII_2$ assembly was modulated by carbon source/metabolism. We used BN-PAGE to monitor $CIII_2$ levels in U2OS cells grown in media containing glucose (*Figure 4—figure supplement 3*). Similar to galactose media, $CIII_2$ abundance was markedly reduced in mitochondria from OCIAD1 knockdown cells grown in glucose media, but not in OCIAD2 knockdown cells, indicating that OCIAD1, but not OCIAD2, affects the assembly of Complex III under our experimental conditions.

To gain insight into OCIAD1 and OCIAD2 function, we also used untargeted quantitative mass spectrometry to compare the whole-cell proteomes of control U2OS cells, U2OS cells with individual or double OCIAD1/OCIAD2 knockdown, and OCIAD1 knockdown U2OS cells in which OCIAD1 expression was reintroduced by lentiviral delivery. Overall, the proteome was resilient to loss of OCIAD1 and OCIAD2 expression, as only 40 proteins were significantly affected in at least one of the different cell lines (*Figure 4—source data 2*). As expected, OCIAD1 and OCIAD2 were significantly downregulated in the individual and double knockdown cell lines, while GFP was only observed in OCIAD1 knockdown cell line in which OCIAD1 expression was reintroduced by lentiviral transduction using GFP as a selection marker. Hierarchical clustering of significantly affected proteins identified a small cluster tightly associated with OCIAD1 (*Figure 4—figure supplement 4A*, red box), containing 4 of the 10 subunits of Complex III, including UQCRC1, UQCRC2, UQCRB, and CYC1, as well as COX7A2L, which regulates Complex III biogenesis by promoting the assembly of $CIII_2$ with CIV to form the $CIII_2CIV$ supercomplex (*Cogliati et al., 2016*; *Lapuente-Brun et al., 2013*; *Lobo-Jarne et al., 2018*). All proteins in this cluster were selectively downregulated in the individual OCIAD1 knockdown and OCIAD1/OCIAD2 double knockdown cell lines, but unaffected in the OCIAD1 rescued cell line and in OCIAD2 knockdown cells (*Figure 4—figure supplement 4A*). We validated these observations using Western blotting and showed that steady-state levels of UQCRC1, UQCRC2, and CYC1 were reduced in mitochondria isolated from OCIAD1 knockdown cells, but not OCIAD2 knockdown cells (*Figure 4—figure supplement 4B* and *Figure 5E,F*).

Although OCIAD2 does not have a measurable effect on Complex III biogenesis under our conditions, BN-PAGE analysis and Western analysis of mitochondria from U2OS cells using anti-OCIAD2 antibody demonstrated that, like OCIAD1, OCIAD2 co-migrates with prohibitin complexes independent of OCIAD1 (*Figure 4—figure supplement 1E*). Reciprocally, OCIAD2 is not required for the association of OCIAD1 with prohibitin complexes as OCIAD1 migrated as 0.9–1 MDa assemblies in K562 cells that do not express OCIAD2 (*Figure 4C*). Therefore, our data suggest that OCIAD1 and OCIAD2 paralogs experienced functional diversification during evolution. However, as OCIAD1 and OCIAD2 have been reported to interact (*Sinha et al., 2018*), we cannot exclude the possibility that OCIAD2 may modulate the role of OCIAD1 in Complex III regulation in a context-dependent manner.

## OCIAD1 is required for the processing of CYC1

To determine the functional significance of OCIAD1 interactors, we asked which interactions were modulated by OCIAD1 loss-of-function in Complex III assembly. To identify OCIAD1 loss-of-function alleles, we performed a structure-function analysis by initially creating tiled deletions along the C-terminus of OCIAD1 and identified a segment of the conserved OCIA domain essential for OCIAD1 function in $CIII_2$ assembly (*Figure 4—figure supplement 5A,B*). Sequence alignments of OCIAD1 genes from distant phylogenetic species identified a highly conserved phenylalanine residue (F102)

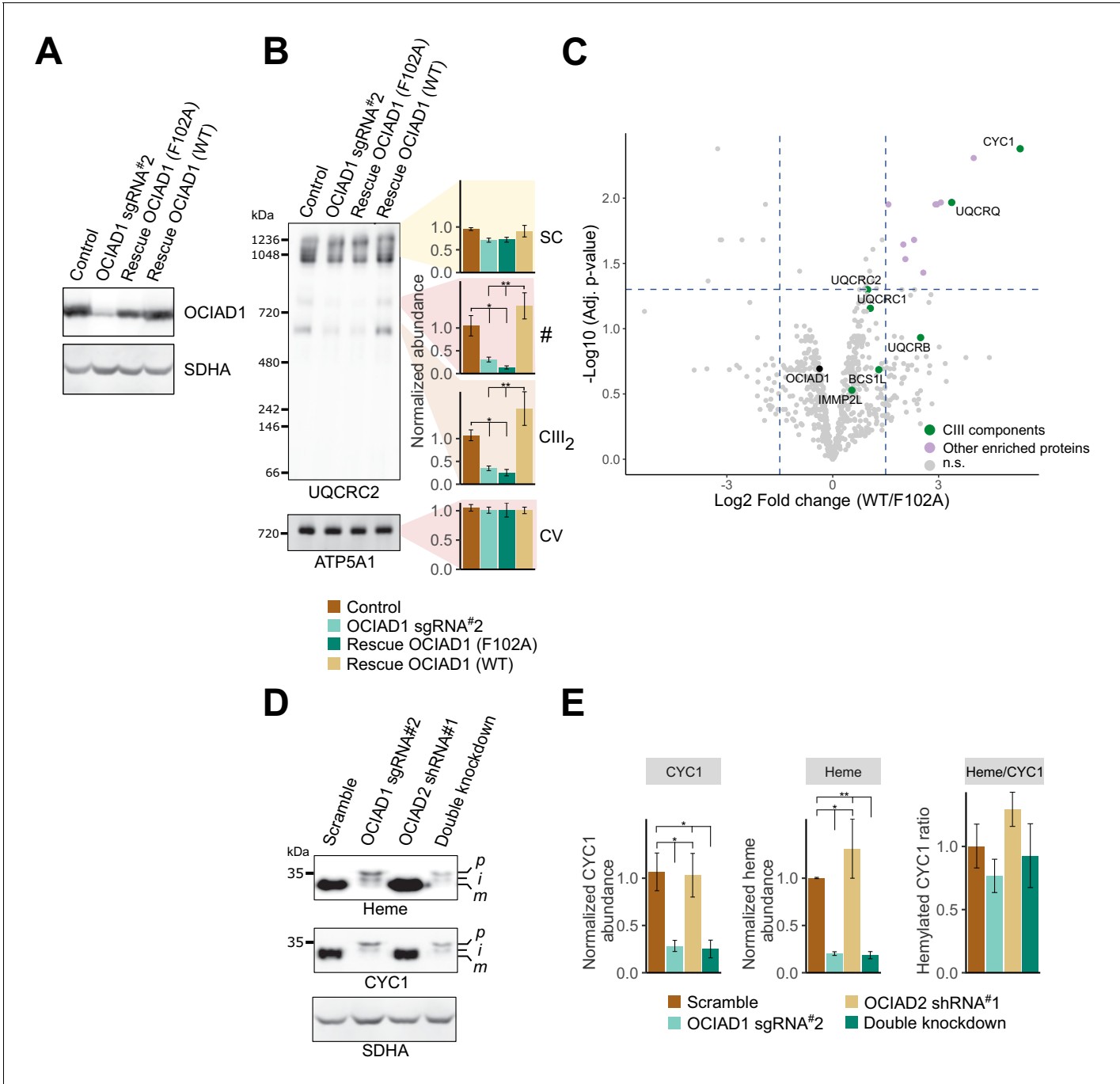

**Figure 5.** Ovarian carcinoma immunoreactive antigen domain-containing protein 1 (OCIAD1) regulates the maturation of cytochrome $c_1$ (CYC1). (**A**) Western blot showing OCIAD1 expression levels in K562 OCIAD1 knockdown cells rescued with either wildtype OCIAD1 or mutant (F102A) OCIAD1. (**B**) Blue-native polyacrylamide gel electrophoresis (PAGE) analysis showing that the F102A point mutant fails to rescue the $CIII_2$ assembly defect. (**C**) Volcano plot showing proteins enriched in OCIAD1 pull-down performed in DSP-crosslinked cell lysate from K562 OCIAD1 knockdown cells rescued with either wildtype or F102A OCIAD1. Proteins with a $log_2$ fold change $\geq 1.5$ and an adjusted p-value<0.05 were considered significantly enriched (n = 3 biological replicates, n.s. = non-significantly enriched). (**D**) Western blot analysis of U2OS mitochondrial membranes solubilized in digitonin. Heme was detected by chemiluminescence before immunoblotting the membrane with the indicated antibodies. (**E**) Quantification of CYC1 (left) and heme (middle) levels from blot shown in (**E**). Right panel shows the proportion of CYC1 that is hemylated. Values represent normalized intensity ± SEM (n = 3 biological replicates). Asterisks (*p<0.05, **p<0.01, or ***p<0.001) correspond to the adjusted (false discovery rate [FDR]) p-values from the post-ANOVA (analysis of variance) pairwise t-test.

The online version of this article includes the following source data and figure supplement(s) for figure 5:

*Figure 5 continued on next page*

Figure 5 continued

**Source data 1.** Output of the limma statistical analysis of the ovarian carcinoma immunoreactive antigen domain-containing protein 1 (OCIAD1) interactome in K562 OCIAD1 knockdown cells rescued with wildtype OCIAD1 or the OCIAD1 F102A mutant.

**Figure supplement 1.** Mature $CIII_2$ contains hemylated cytochrome $c_1$ (CYC1) in ovarian carcinoma immunoreactive antigen domain-containing protein 1 (OCIAD1) knockdown cells.

within this region (*Figure 4—figure supplement 5C*, red box), which we mutated to an alanine. Western blot analysis of mitochondria isolated from K562 OCIAD1 knockdown cells exogenously expressing either wildtype or mutated (F102A) OCIAD1 indicated that OCIAD1 F102A was expressed at near endogenous levels in rescued cells (*Figure 5A*). We examined Complex III assembly by BN-PAGE analysis of mitochondria isolated from K562 OCIAD1 knockdown cells rescued with either wildtype or OCIAD1 F102A. In contrast to cells expressing wildtype OCIAD1, cells expressing the F102A mutant had decreased levels of Complex III, indicating that F102A constitutes a loss-of-function mutation (*Figure 5B*).

We compared the interactomes of wildtype OCIAD1 and OCIAD1 F102A using AE-MS. OCIAD1 was immunopurified from DSP-crosslinked whole-cell lysates prepared from OCIAD1 knockdown cells rescued with either wildtype OCIAD1 or OCIAD1 F102A and analyzed by label-free quantitative mass spectrometry, as shown in *Figure 4A*. This analysis revealed a selective enrichment of CYC1, one of three catalytic $CIII_2$ subunits, in cells rescued with wildtype OCIAD1 versus OCIAD1 F102A *Figure 5C* (*Figure 5—source data 1*). These data suggest that the function of OCIAD1 in Complex III assembly is dependent on its interaction with CYC1.

CYC1 contains a single covalently attached heme prosthetic group that facilitates the transfer of electrons from the Rieske iron–sulfur protein to cytochrome *c*. It is synthesized on cytosolic ribosomes as an apoenzyme precursor with a bipartite signal sequence that is processed in two steps during import. The CYC1 precursor is first processed to an intermediate form by the matrix metalloprotease (MPP), which removes the matrix targeting sequence (*Gasser et al., 1982*; *Ndi et al., 2018*; *Nicholson et al., 1989*). The matrix targeting sequence is followed by a stretch of hydrophobic residues that functions as a signal that stops the translocation of the mature protein across the inner membrane, allowing CYC1 to localize to the IMS. The stop transfer signal is processed by IMMP2L, a signal peptidase-like protease (*Gasser et al., 1982*; *Ndi et al., 2018*; *Nicholson et al., 1989*; *Nunnari et al., 1993*). IMMP2L processing requires the covalent addition of a heme moiety to CYC1, catalyzed by holocytochrome c-type synthase (HCCS), and completes the formation of mature holo-CYC1 (*Ndi et al., 2018*; *Nicholson et al., 1989*).

IMMP2L was identified in our OCIAD1 interactome analysis (*Figure 4A*) and also in a prohibitin interactome analysis (*Yoshinaka et al., 2019*). Therefore, we assessed whether OCIAD1 regulates CYC1 maturation using Western blotting analysis of digitonin-solubilized mitochondria. CYC1 levels were significantly reduced in OCIAD1 knockdown cells (*Figure 5D and E*), consistent with our unbiased mass spectrometry data (*Figure 4—figure supplement 4A*). In addition, two larger molecular weight CYC1 species, likely corresponding to the precursor and intermediate forms, accumulated in OCIAD1 knockdown cells as compared to control cells (*Figure 5D*). This phenotype was also observed in OCIAD1 knockdown cells expressing the OCIAD1 loss-of-function truncation (Δ97–115), which failed to rescue $CIII_2$ assembly (*Figure 5—figure supplement 1A*). These data indicate that OCIAD1 function is required for normal CYC1 processing. Given that hemylation is required for IMMP2L processing of CYC1 (*Nicholson et al., 1989*), we also examined whether CYC1 hemylation was dependent on OCIAD1 function. As CYC1 contains a covalently linked heme moiety, we directly assessed CYC1 hemylation by chemiluminescence of mitochondrial fractions analyzed by SDS-PAGE as previously described (*Dorward, 1993*; *Feissner et al., 2003*). The slower migrating CYC1 species that accumulate in OCIAD1 knockdown cells and OCIAD1 Δ97–115 truncated cells were fully hemylated (*Figure 5D,E*, *Figure 5—figure supplement 1A*). The hemylation levels of the slower-migrating CYC1 species in OCIAD1 knockdown cells were proportional to CYC1 abundance, indicating that OCIAD1 knockdown cells have a CYC1 hemylation ratio comparable to mature CYC1 in control cells (*Figure 5E*). These data indicate that OCIAD1 is not required for CYC1 hemylation. The CYC1 maturation defect was not detected in OCIAD2 knockdown cells and, thus, was specific to loss of OCIAD1 function (*Figure 5D and E*). This is consistent with our results showing that OCIAD2 was not required for $CIII_2$ assembly and the conclusion that OCIAD1 and OCIAD2 are functionally

divergent (*Figure 4—figure supplements 2* and *3*). The processing of GPD2 and AIF, two additional substrates of IMMP2L (*Lu et al., 2008*; *Yuan et al., 2018*), was unaffected in OCIAD1 knockdown cells as assessed by immunoblotting, suggesting that the maturation defect is specific to CYC1 (*Figure 5—figure supplement 1D,E*).

To further investigate the role of OCIAD1 in CYC1 processing, we examined the status of CYC1 in CIII$_2$ in OCIAD1 knockdown cells. We measured the hemylation efficiency of CYC1 in CIII$_2$ resolved by native PAGE (*Figure 5—figure supplement 1B*). Although CIII$_2$ levels were reduced in OCIAD1 knockdown cells, the extent of hemylation in CIII$_2$ was comparable to that of wildtype cells (compare *Figure 5E* and *Figure 5—figure supplement 1C*). We also used 2D-native/SDS-PAGE and found that CIII$_2$ from OCIAD1 knockdown cells contains higher molecular weight CYC1 species (*Figure 6A*). Thus, unprocessed but hemylated CYC1 is incorporated in CIII$_2$ in OCIAD1 knockdown cells.

To better characterize the nature of the CYC1 processivity defect in CIII$_2$, we identified CYC1 peptides using mass spectrometry analysis of BN-PAGE gel slices containing CIII$_2$ assemblies ranging from ~600 to 900 kDa excised from control cells, OCIAD1 knockdown cells, and OCIAD1 knockdown cells rescued with wildtype OCIAD1 (*Figure 6B*). An internal peptide from mature CYC1 (LFDYFPK PYPNSEAAR) was reliably identified in CIII$_2$ assemblies from mitochondria isolated from all cell types, albeit at lower levels in knockdown cells. This is consistent with our whole-cell proteomics and Western blot results showing reduced steady-state levels of CYC1 in OCIAD1 knockdown cells and an overall reduction in CIII$_2$ assemblies by BN-PAGE. We also identified a peptide (TPQAVALSSK), N-terminal to the CYC1 hydrophobic bipartite sequence, that was uniquely detected in CIII$_2$ assemblies isolated from OCIAD1 knockdown cells (*Figure 6B*, lower panel), consistent with the accumulation of the precursor form of CYC1 in OCIAD1 knockdown cells. Conversely, a peptide (SDLELH PPSYPWSHR) representing the N-terminus of mature CYC1, as determined by N-terminome sequencing analysis of the human mitochondrial proteome (*Vaca Jacome et al., 2015*), was only reliably identified in CIII$_2$ assemblies from control and rescued cells, but not from OCIAD1 knockdown cells (*Figure 6B*, middle panel). This peptide is not preceded by an arginine or lysine residue and thus was not produced by tryptic digestion of the CYC1 precursor. Therefore, this peptide distinctly identifies the N-terminus of the mature version of CYC1 (*Vaca Jacome et al., 2015*). Taken together, our results are consistent with OCIAD1 regulating the proteolytic processing and maturation of the holo-CYC1 precursor.

## Discussion

Our data indicate that OCIAD1 is a conserved regulatory determinant of CIII$_2$ assembly that controls the proteolytic processing of holo-CYC1. Cytochrome *bc1* complexes are highly conserved, found in photosynthetic and respiring bacterial plasma membranes of phylogenetically distant species, as well as in eukaryotic cells in mitochondria and in chloroplasts as the related cytochrome b6f complex (*Trumpower, 1990*). The comparison of high-resolution atomic models of cytochrome *bc1* complexes in plants, fungi, and mammals revealed that despite their modest sequence homology, they exist as dimers (CIII$_2$) displaying exceptional structural conservation of all three catalytic subunits (*Maldonado et al., 2021*). In all CIII$_2$ atomic models, mature holo-CYC1 possesses one C-terminal transmembrane helix and an N-terminal domain composed of six $\alpha$-helices and two-strand $\beta$-sheet extending in the IMS (*Maldonado et al., 2021*; *Xia et al., 1997*). This topology is achieved via the highly conserved process of CYC1 maturation. CYC1 contains a bipartite targeting signal composed of two sequential N-terminal presequences: a mitochondrial targeting signal (MTS) processed in the matrix by the MPP and a hydrophobic inner membrane sorting domain. Following hemylation of CYC1 by the heme lyase HCCS, the hydrophobic inner membrane sorting domain is processed by IMMP2L, a subunit of the inner membrane signal peptidase complex (*Arnold et al., 1998*; *Nicholson et al., 1989*; *Nunnari et al., 1993*; *Römisch et al., 1987*; *Sadler et al., 1984*; *van Loon et al., 1987*; *Wachter et al., 1992*). The MTS of CYC1 is required for its targeting to mitochondria; however, its removal by MPP is not required for the heme-dependent maturation of CYC1 by IMMP2L. Thus, both precursor and intermediate forms of CYC1 can be hemylated and the bipartite sequence can be cleaved in a single step by the IMMP2L, without the removal of the MTS by the MPP (*Nicholson et al., 1989*). Consistent with this, two fully hemylated CYC1 species of higher molecular weight accumulate in OCIAD1 knockdown and mutant cells and likely represent the

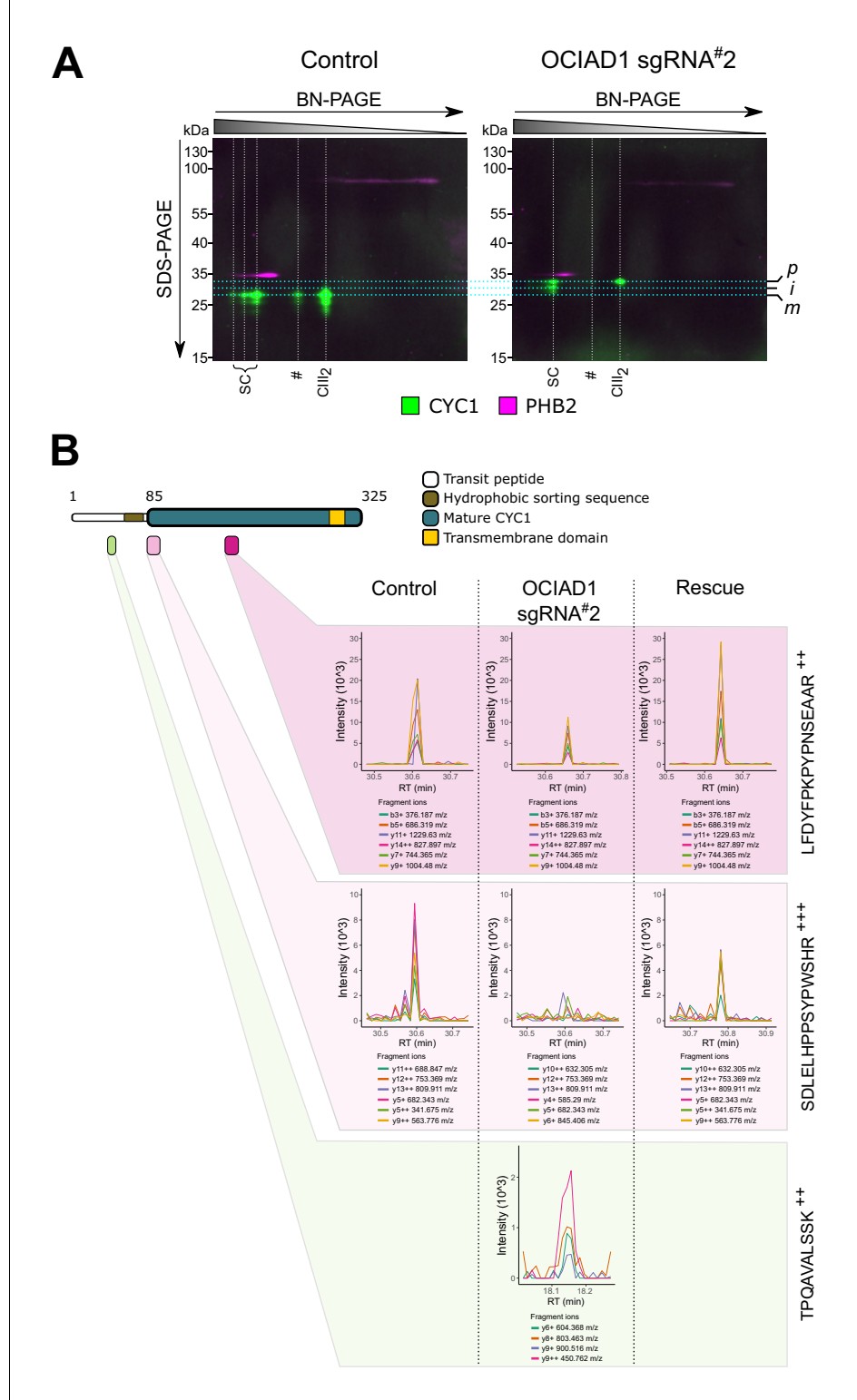

**Figure 6.** Ovarian carcinoma immunoreactive antigen domain-containing protein 1 (OCIAD1) regulates IMMP2L-dependent proteolytic processing of cytochrome $c_1$ (CYC1). (**A**) 2D-native/SDS-PAGE analysis of mitochondrial membranes isolated from K562 control and OCIAD1 knockdown cells and immunoblotted with CYC1 and PHB2 antibodies. CIII$_2$ assemblies from OCIAD1 knockdown cells contained immature CYC1 of higher molecular weight. PHB2 staining served as an internal molecular size reference. Light blue horizontal lines represent the size of putative precursor (p), intermediate (i), and mature (m) CYC1. White vertical lines represent the different high-

*Figure 6 continued on next page*

*Figure 6 continued*

order CIII$_2$ assemblies. (B) Extracted MS2 fragment ion chromatograms (XIC) for three diagnostic CYC1 peptides detected by diaPASEF mass spectrometry in blue-native polyacrylamide gel electrophoresis (BN-PAGE) gel slices excised from control cells, OCIAD1 knockdown cells, and OCIAD1 knockdown cells rescued with wildtype OCIAD1. Individual peptides displayed highly correlated fragment ion co-elution profiles strongly supportive of peptide identification. The TPQAVALSSK$^{++}$ peptide (bottom panel), located at the N-terminus of the CYC1 hydrophobic sorting sequence, was only identified in CIII$_2$ assemblies from OCIAD1 knockdown cells. Conversely, the SDLELHPPSYPWSHR$^{+++}$ peptide (middle panel), which uniquely identifies the N-terminus of mature CYC1 but is not present in the tryptic digest of the CYC1 precursor, was reliably detected in CIII$_2$ assemblies from control and OCIAD1 knockdown cells rescued with wildtype OCIAD1, but not from OCIAD1 knockdown cells. An internal peptide (LFDYFPKPYPNSEAAR$^{+++}$, top panel) common to all CYC1 species (precursor, intermediate, mature) was detected in all cell lines, albeit at lower levels in OCIAD1 cells as expected.

precursor and intermediate holo-CYC1. Our OCIAD1 interactome analysis identified IMMP2L (*Figure 4A*), which is required for the second processing step of CYC1. IMMP2L was also identified in a prohibitin interactome analysis (*Yoshinaka et al., 2019*). Thus, overall, our results suggest that OCIAD1 regulates CYC1 processing by both MPP and IMMP2L.

OCIAD1 and its related paralog OCIAD2 are highly conserved in metazoans. In addition, remote protein homology detection using HHpred analysis found homology similarity between OCIAD paralogs and several yeast proteins involved in respiratory complex formation (*Supplementary file 2*). Yil077c (Rci37), a yeast protein shown to interact with the CIII–CIV supercomplex, m-AAA proteases, and the prohibitin complex (*Morgenstern et al., 2017*), was identified as a putative remote homolog of both OCIAD paralogs. The yeast protein COX20 was also identified as a highly probable homolog of OCIAD1, but not OCIAD2, and has a similar size and topology to OCIAD1. Both proteins are integral inner membrane proteins with N- and C-termini located in the IMS (*Tzagoloff et al., 2000*). COX20 is a chaperone involved in the biogenesis of CIV, where it binds to newly synthesized mitochondrially encoded Cox2 and presents it to the inner membrane peptidase complex to facilitate the proteolytic removal of its N-terminal presequence by Imp2 (*Elliott et al., 2012*; *Nunnari et al., 1993*; *Tzagoloff et al., 2000*). The high homology between COX20 and OCIAD1, but not OCIAD2, is consistent with the functional specialization of OCIAD paralogs and suggests that OCIAD1 serves a conserved function in facilitating the proteolytic processing of CYC1.

Processing of CYC1 to its mature form is not essential for its function in CIII$_2$ as mitochondrial respiration, including CIII$_2$ activity, is not affected in IMMP2L mutant mice (*Bharadwaj et al., 2014*; *Lu et al., 2008*). Consistent with this, we found that immature holo-CYC1 can be successfully incorporated into CIII$_2$ and CIII$_2$-containing supercomplexes in OCIAD1 knockdown and mutant cells. In this context, the functional significance of proteolytic processing of CYC1 is unclear, although it is an evolutionary conserved process. It is possible that CIII$_2$ complexes containing immature holo-CYC1 have increased superoxide production, which can be detrimental to mitochondrial function (*Lu et al., 2008*). However, in contrast to IMMP2L mutant mice (*Bharadwaj et al., 2014*; *Lu et al., 2008*), steady-state levels of CYC1 were also substantially reduced in OCIAD1 knockdown cells, consistent with the observed antimycin A sensitization growth phenotype of OCIAD1 knockdown cells. This observation suggests that OCIAD1 may also function as a chaperone that stabilizes newly imported CYC1.

We demonstrate that under native conditions, a majority of OCIAD1 associates with prohibitins to form supramolecular complexes of ~1 MDa, consistent with published prohibitin interactomes (*Richter-Dennerlein et al., 2014*; *Yoshinaka et al., 2019*). OCIAD1 is likely a direct prohibitin interactor given that residue-to-residue contacts were identified between prohibitin and OCIAD1 peptides using cross-linking mass spectrometry analysis (*Liu et al., 2018*; *Yoshinaka et al., 2019*). Prohibitins are members of the SPFH superfamily of scaffold proteins and form large ring-like structures in membranes, which are thought to create functionally specialized protein and lipid domains within the crowded environment of the IMM (*Osman et al., 2009*). Consistent with this model, prohibitins and the related SPFH scaffold, SLP2, have been shown to sequester inner membrane associated proteases to gate their access to proteolytic substrates (*Merkwirth et al., 2008*; *Steglich et al., 1999*). IMMP2L was identified in the prohibitin interactome by proximity labeling (*Yoshinaka et al., 2019*). Thus, we propose that within prohibitin assemblies, OCIAD1 targets

precursor CYC1 to the IMMP2L peptidase. However, further studies are needed to elucidate the functional significance of the OCIAD1 and prohibitin association in the maturation of CYC1.

# Materials and methods

## Key resources table

| Reagent type (species) or resource | Designation | Source or reference | Identifiers | Additional information |
|---|---|---|---|---|
| Cell line (*Homo sapiens*) | K562 dCas9-KRAB | *Gilbert et al., 2014* doi:10.1016/j.cell.2014.09.029 | | |
| Cell line (*Homo sapiens*) | K562 control | This paper | | K562 dCas9-KRAB transduced with pControlsgRNA |
| Cell line (*Homo sapiens*) | K562 OCIAD1 sgRNA#2 | This paper | | K562 dCas9-KRAB transduced with pOCIAD1sgRNA2 |
| Cell line (*Homo sapiens*) | K562 OCIAD1 rescue | This paper | | K562 OCIAD1 sgRNA#2 transduced with pUltra-OCIAD1 |
| Cell line (*Homo sapiens*) | K562 OCIAD1 (F102A) rescue | This paper | | K562 OCIAD1 sgRNA#2 transduced with pUltra-OCIAD1(F102A) |
| Cell line (*Homo sapiens*) | U2OS dCas9-KRAB | This paper | | U2OS transduced with pMH0006 (Addgene, cat# 135448; *Chen et al., 2019*) |
| Cell line (*Homo sapiens*) | U2OS control | This paper | | U2OS dCas9-KRAB transduced with pControlsgRNA |
| Cell line (*Homo sapiens*) | U2OS OCIAD1 sgRNA #2 | This paper | | U2OS dCas9-KRAB transduced with pOCIAD1sgRNA2 |
| Cell line (*Homo sapiens*) | U2OS OCIAD1 rescue | This paper | | U2OS OCIAD1 sgRNA#2 transduced with pUltra-OCIAD1 |
| Cell line (*Homo sapiens*) | U2OS scramble | This paper | | U2OS control cells transduced with pLKO.1-blast-Scramble (Addgene, cat# 26701) |
| Cell line (*Homo sapiens*) | U2OS OCIAD2 shRNA#1 | This paper | | U2OS control cells transduced with pLKO1-OCIAD2_shRNA1 |
| Cell line (*Homo sapiens*) | U2OS double knockdown | This paper | | U2OS OCIAD1 sgRNA#2 transduced with pLKO1-OCIAD2_shRNA1 |
| Antibody | rabbit polyclonal anti-OCIAD1 | Invitrogen | PA5-20834 | RRID:AB_11155625 (1:2000-1:5000) |
| Antibody | mouse monoclonal anti-OCIAD1 | Proteintech | 66698–1-Ig | RRID:AB_2882051 (1:5000) |
| Antibody | Rabbit polyclonal anti-OCIAD2 | Invitrogen | PA5-59375 | RRID:AB_2644946 (1:500-1:5000) |

*Continued on next page*

*Continued*

| Reagent type (species) or resource | Designation | Source or reference | Identifiers | Additional information |
|---|---|---|---|---|
| Antibody | Mouse monoclonal anti-ATP5A1 | Proteintech | 66037–1-Ig | RRID:AB_11044196 (1:2000-1:5000) |
| Antibody | Rabbit polyclonal anti-NDUFB8 | Proteintech | 14794–1-AP | RRID:AB_2150970 (1:2000) |
| Antibody | Mouse monoclonal anti-SDHA | SantaCruz Biotechnology | sc-166947 | RRID:AB_10610526 (1:2000-1:5000) |
| Antibody | Rabbit polyclonal anti-UQCRC2 | Proteintech | 14742–1-AP | RRID:AB_2241442 (1:2000-1:5000) |
| Antibody | Mouse monoclonal anti-UQCRC1 | Invitrogen | 459140 | RRID:AB_2532227 (1:2000) |
| Antibody | Rabbit polyclonal anti-CYC1 | Proteintech | 10242–1-AP | RRID:AB_2090144 (1:1000) |
| Antibody | Mouse monoclonal anti-COXIV | Proteintech | 66110–1-1g | RRID:AB_2881509 (1:2000) |
| Antibody | Mouse monoclonal anti-PHB2 | Proteintech | 66424–1-Ig | RRID:AB_2811041 (1:5000) |
| Antibody | Rabbit polyclonal anti-TIM50 | Proteintech | 22229–1-AP | RRID:AB_2879039 (1:1000) |
| Antibody | Rabbit polyclonal anti-TOM70 | Proteintech | 14528–1-AP | RRID:AB_2303727 (1:1000) |
| Antibody | Mouse monoclonal anti-GFP | Proteintech | 66002–1-Ig | RRID:AB_11182611 (1:2000) |
| Antibody | Mouse monoclonal anti-β-actin | Proteintech | 66009–1-1g | RRID:AB_2687938 (1:10000) |
| Antibody | Rabbit polyclonal anti-GPD2 | Proteintech | 17219–1-AP | RRID:AB_2112476 (1:1000) |
| Antibody | Rabbit polyclonal anti-AIF | Proteintech | 17984–1-AP | RRID:AB_2224539 (1:1000) |
| Commercial assay or kit | SuperSignal West Femto | Thermo Scientific | 34094 | |
| Chemical compound, drug | Digitonin | Calbiochem | 300410 | |
| Chemical compound, drug | Lauryl maltose neopentyl glycol (LMNG) | Anatrace | NG310 | |
| Chemical compound, drug | Dithiobis (succinimidyl propionate) (DSP) | Life Technologies | 22585 | |
| Chemical compound, drug | LysC/Trypsin | Promega | V5071 | |
| Chemical compound, drug | ProteaseMAX | Promega | V2071 | |
| Other | µMACS protein A beads | Miltenyi Biotec | 130-071-001 | |
| Other | µ Columns | Miltenyi Biotec | 130-042-701 | |
| Other | µMACS Separator | Miltenyi Biotec | 130-042-602 | |

*Continued on next page*

*Continued*

| Reagent type (species) or resource | Designation | Source or reference | Identifiers | Additional information |
|---|---|---|---|---|
| Other | ZipTip with 0.6 µL C18 resin | Millipore Sigma | ZTC18S096 | |
| Other | Disposable gel cutter grids | The Gel Company | MEE2-7-25 | |
| Software, algorithm | DIA-NN | | Version 1.7.12 and 1.7.13 (beta 1) | https://github.com/vdemichev/DiaNN |
| Software, algorithm | Screen Processing | | | https://github.com/mhorlbeck/ScreenProcessing |

## Cell culture

K562 cells and derivatives were cultured in 'RPMI +glucose' (RPMI 1640 from HyClone [cat# SH30255F] or Gibco [cat# 72400047] supplemented with 10% fetal bovine serum [FBS], 100 units/mL penicillin, and 100 µg/mL streptomycin) or glucose-free 'RPMI +galactose' (RPMI 1640 from Gibco [cat# 11879020] supplemented with 10 mM galactose, 25 mM HEPES, 10% FBS, 100 units/mL penicillin, 100 µg/mL streptomycin) where indicated. U2OS cells and derivatives, as well as HEK293T cells, were cultured in 'DMEM + glucose' (DMEM from Gibco [cat# 12430054] supplemented with 10% FBS, 100 units/mL penicillin, 100 µg/mL streptomycin) or glucose-free 'DMEM + galactose' (DMEM from Gibco [cat# 11966025] supplemented with 10 mM galactose, 10% FBS, 100 units/mL penicillin, 100 µg/mL streptomycin) where indicated. Cell lines tested negative for mycoplasma contamination using ATCC universal mycoplasma detection kit (cat# 30-1012K).

## Cloning and plasmid construction

Sequences of oligonucleotides used for cloning are provided in *Supplementary file 3*. Cloning was performed using Phusion or Platinum SuperFi high fidelity DNA polymerases (Thermo Scientific, cat# F530S and 12351010) and Gibson assembly master mix (New England BioLabs, cat# E2611). Individual OCIAD1 sgRNA and OCIAD2 shRNA vectors were generated by annealed oligo cloning of top and bottom oligonucleotides (Integrated DNA Technologies, Coralville, IA) into an optimized lentiviral pU6-sgRNA Ef1α-Puro-T2A-BFP vector digested with BstXI/BlpI (Addgene, cat# 84832) and a pLKO.1 backbone digested with AgeI/EcoRI (Addgene, cat# 26655), respectively. OCIAD1 was initially cloned from human cDNA into a pAcGFP-N1 vector (Clontech, Mountain View, CA). The $GFP_{1-10}$ vectors were cloned by Gibson assembly into an FUGW lentiviral backbone (Addgene, cat# 14883) digested with BamHI/EcoRI. The MTS- and IMS-targeting sequences were ordered as gene blocks (Integrated DNA Technologies, Coralville, IA) and the $GFP_{1-10}$ fragment was cloned from a $pCMV-mGFP_{1-10}$ plasmid (*Van Engelenburg and Palmer, 2010*). The MTS from yeast COX4 (a.a. 1–21) (*Friedman et al., 2011*) and the IMS-targeting signal from MICU1 (a.a. 1–60) (*Gottschalk et al., 2019*; *Hung et al., 2014*; *Tsai et al., 2016*) were chosen to target $GFP_{1-10}$ to the matrix or IMS, respectively. The $pGFP_{11}$-N1 and $pGFP_{11}$-C1 vectors were cloned by Gibson assembly into a pEGFP-N1 backbone (Clontech, Mountain View, CA) digested with BamHI/NotI to replace the GFP gene with the $GFP_{11}$ β-barrel. The $GFP_{11}$ fragments were ordered as gBlocks (Integrated DNA Technologies, Coralville, IA) and contained a strategically located BamHI cloning site for easy N- or C-terminal tagging. CoQ9 and MICU1 genes were cloned from human cDNA and inserted into a BamHI-digested $pGFP_{11}$-N1 vector by Gibson assembly to generate the $pCoQ9-GFP_{11}$ and $pMICU1-GFP_{11}$ plasmids. Similarly, OCIAD1 was amplified from the pAcGFP-OCIAD1 plasmid and cloned into BamHI-digested $pGFP_{11}$-N1 and $pGFP_{11}$-C1 vectors to create the $pOCIAD1-GFP_{11}$ and $pGFP_{11}$-OCIAD1 plasmids, respectively. OCIAD1 was also amplified from the pAcGFP-OCIAD1 plasmid and cloned into a XbaI/BamHI-digested pUltra-EGFP backbone (Addgene, cat# 24129) to generate a lentiviral vector expressing the GFP-OCIAD1 fusion gene containing a 'self-cleaving' P2A sequence. The OCIAD1 F102A point mutant was generated from this pUltra-OCIAD1 vector using site-directed mutagenesis. We also generated an OCIAD1 construct with a C-terminal StrepII tag preceded by a TEV cleavage site. For this, OCIAD1 was amplified from the pAcGFP-OCIAD1 plasmid and inserted

into a XbaI/BamHI-digested pUltra-EGFP vector by Gibson assembly, together with a gBlock (Integrated DNA Technologies, Coralville, IA) encoding the TEV-StrepII sequence. The OCIAD1 truncation constructs were generated by inverse PCR using the pAcGFP-OCIAD1 and pUltra-OCIAD1-TEV-StrepII vectors as templates. Finally, to generate lentiviral vectors expressing the truncated OCIAD1 isoforms with a C-terminal GFP tag, the entire OCIAD1-GFP cassette containing the deletion was amplified from the various pAcGFP-OCIAD1 truncated constructs and cloned by Gibson assembly into an FUGW plasmid digested with BamHI/EcoRI to remove its GFP gene.

## Lentivirus production, infection, and generation of cell lines

Lentivirus were generated by transfecting HEK293T cells with standard packaging vectors using TransIT-LT1 Transfection Reagent (Mirus Bio, Madison, WI) or Lipofectamine 2000 (Invitrogen, Carlsbad, CA) according to the manufacturer's instructions. Briefly, HEK293T were plated in a six-wells plate on day 0 ($0.5 \times 10^6$ cells per well) and transfected on day 1 with a liposome/DNA mixture containing the following packaging plasmids (0.1 µg of pGag/Pol, 0.1 µg of pREV, 0.1 µg of pTAT, and 0.2 µg of pVSVG) and 1.5 µg of lentiviral vector. On days 3 and 4, the media was replenished with 3 mL of fresh DMEM + glucose media. On days 4 and 5, the viral suspensions were harvested, pooled, pelleted at 1000 g for 5 min, and the supernatant was filtered through 0.45 µm PES filters (Thermo Scientific, cat# 725–2545). The viral suspension was either used directly or kept frozen at −80℃ until transduction. For transduction, U2OS and K562 cells were plated in six-well plates (175,000 and 200,000 cells/well, respectively) and infected with 0.5–2 mL of viral suspension supplemented with polybrene at a final concentration of 8 µg/mL. Infected cells were grown for several days before selection with antibiotics or FACS.

K562 dCas9-KRAB cells were previously published (*Gilbert et al., 2013*) and authenticated by ATCC using STR profiling. U2OS dCas9-KRAB cells were generated by lentiviral transduction of U2OS cells obtained from ATCC (HTB-96) with pMH0006 (Addgene, cat# 135448; *Chen et al., 2019*) and selected for BFP expression by FACS. CRISPRi knockdown and control cell lines were generated by subsequent lentiviral transduction of dCas9 lines with plasmids containing individual sgRNAs (pOCIAD1sgRNA1 or pOCIAD1sgRNA2) or a non-targeting sgRNA and selected for higher levels of BFP expression by FACS. OCIAD1 knockdown cell lines rescued with wildtype or F102A OCIAD1 were generated by lentiviral transduction with plasmids pUltra-OCIAD1 and pUltra-OCIAD1(F102A), respectively, and selected for GFP expression by FACS. The U2OS OCIAD2 shRNA knockdown cells were generated by lentiviral transduction with plasmids containing individual shRNAs and selected with 15 µg/mL blasticidin for 7 days. The U2OS OCIAD1/2 double knockdown cell line was generated by infecting stable U2OS CRISPRi cells stably expressing sgRNA#2 (above) with the lentivirus vector pLKO1-OCIAD2_shRNA1 and selecting infected cells with 15 µg/mL blasticidin for 7 days. A control cell line was generated by infecting U2OS cells stably expressing a non-targeting sgRNA (above) with the lentivirus vector pLKO.1-blast-Scramble (Addgene, cat# 26701) expressing a non-targeting shRNA sequence and selected with 15 µg/mL blasticidin for 7 days. Cell lines expressing truncated OCIAD1 constructs were generated by lentiviral infection of CRISPRi cells stably expressing sgRNA#2 (above) with the indicated pFUGW-OCIAD1 and pUltra-OCIAD1-TEV-StrepII lentiviral vectors. U2OS cells stably expressing matrix- or IMS-targeted $GFP_{1-10}$ were generated by lentiviral transduction with the plasmids pMTS-GFP1-10 and pIMS-GFP1-10, respectively.

## Genome-scale CRISPRi screening

Genome-scale CRISPRi screens were conducted on two biological replicates as previously described (*Gilbert et al., 2014*; *Horlbeck et al., 2016*; *Jost et al., 2017*). Briefly, K562 cells expressing dCas9-KRAB were transduced with the pooled hCRISPRi-v2 sgRNA library (*Horlbeck et al., 2016*) and selected for 2 days with 0.75 µg/mL puromycin. Cells were then allowed to recover for 2 days in puromycin-free media before freezing library-containing cell aliquots ($150 \times 10^6$ cells per aliquot) under liquid nitrogen. After subsequent expansion and freezing while maintaining equivalent cell numbers, biological replicates were performed from two independent cell aliquots. Upon thawing, cells were recovered in RPMI +glucose for 4 days followed by 6 days conditioning in RPMI + galactose. At this point, $t_0$ samples with a minimum 750× library coverage ($150 \times 10^6$ cells) were harvested while $250 \times 10^6$ cells each were seeded in separate 3 L spinner flasks (500 mL of media at $0.5 \times 10^6$ cells/mL) for treatment. Cells were treated with four pulses of antimycin (3.5–

3.75 nM) or vehicle (ethanol), consisting of 24 hr drug treatment, washout, and 48 hr recovery. For the duration of the screen, cells were maintained in RPMI + galactose at $0.5 \times 10^6$ cells/mL by daily media dilution (minimum daily coverage ~1000 cells per sgRNA). At the end of the screen, endpoint samples from treated and vehicle-treated population ($150 \times 10^6$ cells each) were harvested and frozen. Genomic DNA was isolated from frozen cell pellets at the indicated time points and the sgRNA-encoding region was enriched, amplified, and processed for sequencing on an Illumina HiSeq 4000 platform as described previously (**Horlbeck et al., 2016**).

Sequencing reads were aligned to hCRISPRi-v2 library and counted using the Python-based ScreenProcessing pipeline (https://github.com/mhorlbeck/ScreenProcessing) (**Horlbeck et al., 2016**). Negative control genes were generated and phenotypes and Mann-Whitney p-values were calculated as described previously (**Gilbert et al., 2014**; **Horlbeck et al., 2016**; **Jost et al., 2017**). Briefly, antimycin A sensitivity phenotypes (ρ) were determined by calculating the $\log_2$ fold change in counts of an sgRNA in the treated and untreated samples, subtracting the equivalent median value for all non-targeting sgRNAs, and dividing by the number of population doubling differences between the treated and untreated populations (**Gilbert et al., 2014**; **Jost et al., 2017**; **Kampmann et al., 2013**). Phenotypes from sgRNAs targeting the same gene were collapsed into a single phenotype for each gene using the average of the three sgRNAs with the strongest phenotypes by absolute value and assigned a p-value using the Mann-Whitney test of all sgRNAs targeting the same gene compared to the non-targeting controls. For genes with multiple independent transcription start sites (TSSs) targeted by the sgRNA library, phenotypes and p-values were calculated independently for each TSS and then collapsed to a single score by selecting the TSS with the lowest Mann-Whitney p-value, as described previously (**Gilbert et al., 2014**; **Horlbeck et al., 2016**; **Jost et al., 2017**). Read counts and phenotypes for individual sgRNAs as well as gene-level phenotypes are available in **Supplementary file 1**.

## Validation of individual sgRNA phenotypes

The antimycin screen phenotype was validated by a growth competition assay using K562 cells expressing individually cloned sgRNAs. In short, K562 dCas9-KRAB cells were mixed with an equal number of K562 CRISPRi cells expressing a non-targeting sgRNA or sgRNA against OCIAD1. The sgRNA expression construct expressed a BFP reporter to identify infected cells. Of note, the dCas9-KRAB construct also expressed BFP fused to dCas9, but the BFP fluorescent intensity was dim and sgRNA-infected cells were clearly distinguishable from dCas9 cells by flow cytometry. For simplicity, K562 CRISPRi cells are referred to as BFP+ in the text. Cells were grown for 24 hr in RPMI + galactose containing either antimycin (5 nM) or vehicle (ethanol), washed, and allowed to recover for 72 hr. The proportion of BFP-positive cells in each cell mixture was determined at the indicated time points using an Amnis Imagestream X (Luminex, Austin, TX) flow cytometer.

## Mitochondria isolation

For mitochondria isolation, all procedures were performed on ice or at 4°C. U2OS cells were grown to confluency in 150 mm Petri dishes and washed three times with 15 mL of cold homogenization buffer (10 mM HEPES, 1 mM EDTA, 210 mM mannitol, 70 mM sucrose, pH 7.4 at 4°C). Cells were harvested by scraping in cold homogenization buffer (0.75 mL per plate) supplemented with 1× protease inhibitor cocktail (MilliporeSigma, Burlington, MA) and lysed with 6–8 strokes of a glass Dounce homogenizer fitted with a tight pestle. At this point, a small fraction of homogenate was immediately snap-frozen on liquid nitrogen and stored at −80°C for whole-cell proteomics analysis as described below. For K562 suspension cells, cells were harvested by centrifugation (1000 g, 5 min), washed with cold homogenization buffer, re-pelleted (1000 g, 5 min), and incubated on ice for 20 min in swelling buffer (10 mM HEPES, 1 mM EDTA, pH 7.4 at 4°C) supplemented with 1× protease inhibitor cocktail (MilliporeSigma, Burlington, MA). Cells were then lysed with 25 strokes of a glass Dounce homogenizer fitted with a tight pestle and immediately diluted with 2× homogenization buffer (10 mM HEPES, 1 mM EDTA, 420 mM mannitol, 140 mM sucrose, supplemented with 1× protease inhibitor cocktail, pH 7.4 at 4°C) to a final concentration of 10 mM HEPES, 1 mM EDTA, 210 mM mannitol, 70 mM sucrose. The homogenate was centrifuged at ~1300 g for 5 min to remove nuclei, unbroken cells, and large cellular debris and the supernatant was centrifuged at ~14,000 g for 10 min at 4°C. The crude mitochondrial pellet was resuspended in homogenization buffer

supplemented with 1× protease inhibitor cocktail prior to measuring protein concentration using a bicinchoninic acid (BCA) assay (Pierce, Waltham, MA). Mitochondrial samples were either used immediately or snap-frozen in 50 or 200 µg aliquots on liquid nitrogen and stored at −80°C.

## Native PAGE analysis

BN and clear-native (CN) PAGE analyses were performed as previously described (*Wittig et al., 2007*; *Wittig et al., 2006*). All procedures were performed on ice or at 4°C. Mitochondrial aliquots (200 µg) were thawed on ice, diluted with 1 mL of solubilization buffer (50 mM imidazole, 50 mM NaCl, 2 mM 6-aminohexanoic acid, 1 mM EDTA, pH7.0 at 4°C), and pelleted at 21,300 g for 10 min. The supernatant was removed and the mitochondrial pellet was resuspended in 20 µL of solubilization buffer supplemented with digitonin (Calbiochem, cat# 300410) or LMNG (Anatrace, Maumee, OH) to a final detergent-to-protein ratio of 4 and 1 g/g, respectively. Samples were solubilized on ice for ~15 min and centrifuged at 21,300 g for 20 min. The supernatant was collected and protein concentration was measured using a BCA assay kit (Pierce, Waltham, MA).

For BN-PAGE, solubilized mitochondrial membranes were supplemented with 50% glycerol to a final concentration of 5% and Coomassie blue G-250 dye to a final detergent/dye ratio of 8 g/g. Equivalent amount of proteins were loaded on 3–12% polyacrylamide gels. The electrophoresis was started with cathode buffer B (50 mM tricine, 7.5 mM imidazole, 0.02% Coomassie blue G-250, pH ~7.0) and exchanged with cathode buffer B/10 (50 mM tricine, 7.5 mM imidazole, 0.002% Coomassie blue G-250, pH ~7.0) once the migration front had reached ~1/3 of the resolving gel. For CN-PAGE, the solubilized mitochondrial samples were supplemented with 50% glycerol, 0.1% Ponceau S to a final concentration of ~5% glycerol and ~0.01% Ponceau S. Equivalent amount of proteins were loaded on 3–12% polyacrylamide gels. The cathode buffer contained 50 mM tricine, 7.5 mM imidazole, 0.01% dodecylmaltoside, and 0.05% sodium deoxycholate (pH ~7.0). The composition of the anode buffer (25 mM imidazole, pH 7.0) was the same for BN-PAGE and CN-PAGE and remained constant for the duration of the electrophoresis. Gels were run in a cold room (4°C) at 100 V until the samples had entered the resolving gel and at 275 V thereafter. After electrophoresis, the gels were incubated in denaturing buffer (300 mM Tris, 100 mM acetic acid, 1% SDS, pH 8.6) at room temperature with agitation for 20 min and stored at room temperature between two glass plates for 1 hr to evenly distribute the SDS. Proteins were then electroblotted in at 4°C onto low fluorescent PVDF membranes at 90 mA and a voltage limited to 20 V for 12–14 hr using a wet tank transfer apparatus filled with cold transfer buffer (150 mM Tris, 50 mM acetic acid, pH 8.6). BN-PAGE membranes were partially destained in 25% methanol, 10% acetic acid to visualize the ladder, and completely destained with 100% methanol for Western blotting analysis. CN-PAGE membranes were stained with 5% acetic acid, 0.1% Ponceau S (w/v) to visualize the ladder, and destained completely with extensive water washes before Western blotting analysis.

For 2D-native/SDS-PAGE analysis, individual gel lanes were excised from BN-PAGE gels immediately after electrophoresis and incubated in 8–10 mL of denaturing buffer (62.5 mM Tris pH 6.8, 2% SDS, 10% glycerol, 10 mM TCEP) in a 15 mL Falcon tube for 20 min at room temperature under gentle agitation. The gel strips were then equilibrated in 1× SDS-PAGE running buffer at room temperature for 15 min, loaded horizontally on a 10% polyacrylamide gel, and processed for Western blotting analysis as described below.

For the mobility shift assay, 400 µg of K562 mitochondria was solubilized with LMNG at a 1 g/g ratio as described above. The sample was halved and incubated with either mouse anti-PHB2 antibodies (Proteintech, cat# 66424–1-Ig, 70 ng, ~1.8 µL) or vehicle (PBS) on ice of 90 min. Samples were then analyzed by BN-PAGE as described above.

## Protease protection and carbonate extraction analysis

Protease protection analysis was performed on mitochondria freshly isolated from U2OS cells as previously described (*Hoppins et al., 2011*) with the following modifications. Mitochondria (50 µg of total mitochondrial protein) were resuspended in 500 µL of one of the following solutions: homogenization buffer (210 mM mannitol, 70 mM sucrose, 10 mM HEPES, 1 mM EDTA, pH 7.4), mitoplast/swelling buffer (10 mM HEPES, pH 7.4), or solubilizing buffer (homogenization buffer with 1% Triton X-100). After 15 min incubation on ice, the mitoplast/swelling sample was gently pipetted up and down 15 times to disrupt the OMM. Proteinase K was then added to the indicated samples to a final

concentration of 100 µg/mL, and samples were incubated on ice for 20 min. The digestion was stopped by adding PMSF to a final concentration of 2 mM and incubating the samples on ice for 5 min. TCA was then added to a final concentration of 12.5% and proteins were precipitated on ice for 1 hr. Proteins were then pelleted by centrifugation at 21,130 g for 15 min at 4°C, washed with acetone, dried, and resuspended in 100 µL of 1× Laemmli buffer. Samples (20 µL) were loaded on a 10% SDS-PAGE and analyzed by Western blotting with the indicated antibodies as described below.

The carbonate extraction assay was performed as described (*Hoppins et al., 2011*) with the following modifications. Mitochondria isolated from U2OS cells (50 µg of total mitochondrial protein) were thawed on ice, pelleted at 15,000 g for 10 min at 4°C, and resuspended in 200 µL of one of the following solutions: 10 mM HEPES (pH 7.4), 100 mM sodium carbonate (pH 10.5), 100 mM sodium carbonate (pH 11), or 100 mM sodium carbonate (pH 11.5). Samples were incubated on ice for 30 min and centrifuged at 100,000 g for 1 hr in a TLA100 rotor. The supernatant was harvested and proteins were precipitated with TCA as described above. The pellet fraction and TCA-precipitated proteins were resuspended in 50 µL of 1× Laemmli buffer and 10 µL was loaded on a 10% SDS-PAGE and analyzed by Western blotting with the indicated antibodies as described below.

## Western blotting analysis

For quantitative Western blot analysis, protein concentration was determined using a BCA assay kit (Pierce, Waltham, MA) and equivalent amount of proteins were diluted with 6× Laemmli sample buffer to a final concentration of 62.5 mM Tris pH 6.8, 2% SDS, 10% glycerol, 0.1M DTT, 0.01% bromophenol blue. Samples were heated for 2–5 min at 95°C and loaded on 10% Tris-glycine polyacrylamide gels. After electrophoresis, proteins were electroblotted on low fluorescent PVDF or nitrocellulose membranes, and immunoblotted with the following primary antibodies: rabbit anti-OCIAD1 (Invitrogen, cat# PA5-20834, 1:2000–1:5000), mouse anti-OCIAD1 (Proteintech, cat# 66698–1-Ig, 1:5000), rabbit anti-OCIAD2 (Invitrogen, cat# PA5-59375, 1:500–1:5000), mouse anti-ATP5A1 (Proteintech, cat# 66037–1-Ig, 1:2000–1:5000), rabbit anti-NDUFB8 (Proteintech, cat# 14794–1-AP, 1:2000), mouse anti-SDHA (SantaCruz Biotechnology, cat# sc-166947, 1:2000–1:5000), rabbit anti-UQCRC2 (Proteintech, cat# 14742–1-AP, 1:2000–1:5000), mouse anti-UQCRC1 (Invitrogen, cat# 459140, 1:2000), rabbit anti-CYC1 (Proteintech, cat# 10242–1-AP, 1:1000), mouse anti-COXIV (Proteintech, cat# 66110–1-1g, 1:2000), mouse anti-PHB2 (Proteintech, cat# 66424–1-Ig, 1:5000), rabbit anti-TIM50 (Proteintech, cat# 22229–1-AP, 1:1000), rabbit anti-TOM70 (Proteintech, cat# 14528–1-AP, 1:1000), mouse anti-GFP (Proteintech, cat# 66002–1-Ig, 1:2000), mouse anti-β-actin (Proteintech, cat# 66009–1-1g, 1:10,000), rabbit anti-GPD2 (Proteintech, cat# 17219–1-AP, 1:1000), rabbit anti-AIF (Proteintech, cat# 17984–1-AP, 1:4000). Secondary antibodies conjugated to DyLight 680 and DyLight 800 (Thermo Fisher Scientific, 1:5000) were used and visualized with an Odyssey Infrared Imaging System (LI-COR, Lincoln, NE). Densitometry analysis was done using the quantification software ImageStudio Lite (LI-COR, Lincoln, NE).

## Heme detection

Chemiluminescence was used to detect c-type heme on PVDF or nitrocellulose membranes as previously described (*Dorward, 1993*; *Feissner et al., 2003*). In short, membranes were rinsed with distilled water immediately after electrophoresis, incubated with SuperSignal West Femto chemiluminescent substrate (Pierce, Waltham, MA), and imaged on an ImageQuant LAS 4000 (GE, Boston, MA). Densitometry analysis was done using the quantification software ImageStudio Lite (LI-COR, Lincoln, NE).

## GFP complementation assay

U2OS cells stably expressing $GFP_{1–10}$ in the matrix (MTS) or IMS were plated in six-well plate (~300,000 cells/well) on day 0 and transfected on day 1 with 6 µL of Lipofectamine 2000 (Invitrogen, Carlsbad, CA), according to the manufacturer's instructions. The cells were transfected with 250 ng of the following plasmids: $CoQ9-GFP_{11}$, $GFP_{11}-MICU1$, $OCIAD1-GFP_{11}$, and $GFP_{11}-OCIAD1$, and 750 ng of transfection carrier DNA (Promega, pGEM2 plasmid). Cells were expanded in 10 cm plate on day 2 and analyzed by fluorescent flow cytometry on day 3 with an Amnis Imagestream X (Luminex, Austin, TX).

## Immunopurification

Cells were crosslinked with dithiobis(succinimidyl propionate) (DSP, Life Technologies, cat# 22585) made from a freshly prepared 0.25 M stock solution in DMSO. In short, 150 mL of confluent (~$1\times10^6$ cells/mL) K562 cells of the indicated OCIAD1 background were harvested by centrifugation (1000 g, 5 min), washed with warm (37°C) PBS, and crosslinked at room temperature for 30 min with 0.5 mM DSP in PBS at ~$1\times10^6$ cells/mL. DSP was then quenched by adding Tris-HCl (pH 7.5) to a final concentration of 100 mM. Cells were harvested by centrifugation (1000 g, 5 min), washed with cold PBS, harvested again, and solubilized in 2 mL of cold RIPA buffer supplemented with $1\times$ protease inhibitor cocktail (MilliporeSigma, Burlington, MA) on ice for 30 min. Samples were centrifuged at 26,000 g for 30 min at 4°C in a TLA100.4 rotor. The supernatant was collected, protein concentration was measured using a BCA assay kit (Pierce, Waltham, MA), and aliquots were stored at −80°C.

Immunopurification was performed on three independently DSP-crosslinked samples. Each sample was thawed on ice and adjusted to 7.8 mg of total protein in 2 mL of RIPA buffer containing $1\times$ protease inhibitor cocktail (MilliporeSigma, Burlington, MA). OCIAD1 was immunocaptured overnight at 4°C with 3 µg of rabbit anti-OCIAD1 antibody (Thermo Fisher Scientific, cat# PA5-20834). Antibodies were captured with 100 µL of µMACS protein A beads (Miltenyi Biotec, San Diego, CA). Beads were isolated with µ columns and a µMACS separator (Miltenyi Biotec, San Diego, CA), washed five times with 1 mL of RIPA buffer and three times with 1 mL of 50 mM ammonium bicarbonate pH 8.0. Bait proteins were eluted with 25 µL of elution buffer (2 M urea, 0.67 M thiourea in 50 mM ammonium bicarbonate pH 8.0) containing LysC/Trypsin (Promega, Madison, WI, cat# V5071) to a final concentration of 5 µg/mL followed by two elution with 50 µL of elution buffer without LysC/Trypsin. Samples were reduced with 10 mM TCEP (Pierce, Waltham, MA) for 30 min at 37°C, alkylated with 15 mM 2-chloroacetamide (MilliporeSigma, Burlington, MA), digested overnight at 37°C, and desalted using ZipTip with 0.6 µL C18 resin (MilliporeSigma, Burlington, MA, cat# ZTC18S096) prior to LC-MS/MS analysis as described below.

## Protein digestion on suspension traps

Protein digestion of U2OS lysates was done on suspension traps (S-Trap) as described (*Ludwig et al., 2018*) with the following modifications. Whole-cell and crude mitochondrial lysates (50 µg total protein) were boiled in 5% SDS, 50 mM ammonium bicarbonate (pH 7.55) for 5 min. Proteins were then reduced with 10 mM TCEP for 15 min at 37°C and alkylated in the dark for 30 min with 15 mM 2-chloroacetamide. The proteins were then acidified with phosphoric acid (final concentration of 1.2%) and diluted with six volumes of S-Trap buffer (90% methanol, 100 mM ammonium bicarbonate, pH 7.1). The colloidal suspension was loaded onto DNA miniprep spin columns used as 'suspension traps' (EZ-10 DNA spin columns, Biobasic, Amherst, NY) and washed with S-Trap buffer prior to overnight proteolysis at 37°C with LysC/trypsin (Promega, Madison, WI) in 50 mM ammonium bicarbonate (pH 8.0) at a protease/protein ratio of 1:40 (w/w). Peptides were successively eluted with 40 µL of 50 mM ammonium bicarbonate (pH 8.0), 40 µL of ultrapure Milli-Q water, 0.1% TFA, and 40 µL of 80% acetonitrile, 0.1% TFA in ultrapure Milli-Q water. Peptides were dried using a SpeedVac concentrator and resuspended in 30 µL of 2% acetonitrile, 0.1% TFA. Peptide concentration was measured using a fluorometric peptide assay kit (Pierce, Waltham, MA) and samples were analyzed by LC-MS/MS as described below.

## In-gel protein digestion

To minimize contamination, procedures were performed in a biosafety cabinet whenever possible. Mitochondria from U2OS cells of the indicated OCIAD1 background were solubilized with digitonin at a 4 g/g detergent/protein ratio and 100 µg of solubilized mitochondrial protein was resolved by BN-PAGE as described above. After electrophoresis, the gel was fixed with 40% methanol, 10% acetic acid at room temperature for 20 min and destained with 8% acetic acid for 20 min. Gel slices (2 mm × 7 mm) were excised along the entire lane using disposable gel cutter grids (The Gel Company, San Francisco, CA, cat# MEE2-7-25). Ten gel slices ranging from ~600 to 900 kDa were collected in 100 µL of 50 mM ammonium bicarbonate (pH 8.0) in a 96-well plate and destained/dehydrated with successive 5 min washes with 100 µL of the following solutions (three washes each): 50 mM ammonium bicarbonate (pH 8.0), 25% acetonitrile in 50 mM ammonium bicarbonate (pH 8.0), 50% acetonitrile in 50 mM ammonium bicarbonate (pH 8.0), 75% acetonitrile in 50 mM

ammonium bicarbonate (pH 8.0), 100% acetonitrile. Proteins were then reduced with 50 μL of 10 mM TCEP for 30 min at 37°C, gel slices were dehydrated again with three washes with 100% acetonitrile, and alkylated with 15 mM 2-chloroacetamide in the dark for 20 min. Gel slices were dehydrated again and washed for 5 min with 100 μL of the following solutions (two washes each): 50 mM ammonium bicarbonate (pH 8.0), 25% acetonitrile in 50 mM ammonium bicarbonate (pH 8.0), 50% acetonitrile in 50 mM ammonium bicarbonate (pH 8.0), 75% acetonitrile in 50 mM ammonium bicarbonate (pH 8.0), and four washes with 100% acetonitrile. Gel slices were air-dried before overnight ProteaseMax-aided digestion as previously described (*Saveliev et al., 2013*). In short, dried gel pieces were rehydrated in 50 μL of 12 ng/μL LysC/Trypsin (Promega, Madison, WI), 0.01% ProteaseMAX surfactant (Promega, Madison, WI, cat# V2071) in 50 mM ammonium bicarbonate (pH 8.0) for 20 min on ice and overlaid with 50 μL of 0.01% ProteaseMAX surfactant in 50 mM ammonium bicarbonate (pH 8.0). Proteins were digested overnight at 37°C. The peptide-containing solution was collected in 1.5 mL eppendorf tubes and 100 μL of 75% acetonitrile, 1% TFA in 25 mM ammonium bicarbonate (pH 8.0) was added to each gel slice to elute remaining peptides. Both eluates were pooled and dried using a SpeedVac concentrator before LC-MS/MS analysis as described below.

## Mass spectrometry analysis

LC-MS/MS analysis was performed at the University of California, Davis, Genome Center Proteomics Core. Immunoprecipitation and whole-cell samples were run on a Thermo Scientific Fusion Lumos mass spectrometer in data-independent acquisition (DIA) mode. Peptides were separated on an Easy-spray 100 μm × 25 cm C18 column using a Dionex Ultimate 3000 nUPLC with 0.1% formic acid (solvent A) and 100% acetonitrile, 0.1% formic acid (solvent B) and the following gradient conditions: 2–50% solvent B over 60 min, followed by a 50–99% solvent B in 6 min, held for 3 min and finally 99–2% solvent B in 2 min. The total run time was 90 min. Six gas phase fractionated (GPF) chromatogram library injections were acquired using 4 Da staggered isolation windows (GPF 1: 400–500 m/z, GPF 2: 500–600 m/z, GPF 3: 600–700 m/z, GPF 4: 700–800 m/z, GPF 5: 800–900 m/z, and GPF 6: 900–1000 m/z). Mass spectra were acquired using a collision energy of 35, resolution of 30 K, maximum inject time of 54 ms, and an AGC target of 50 K. The analytical samples were run in DIA mode with 8 Da staggered isolation windows covering 400–1000 m/z.

BN-PAGE gel samples were run on a Bruker TimsTof Pro mass spectrometer. Peptides were directly loaded on a Ionoptiks (Parkville, Victoria, Australia) 75 μm × 25 cm 1.6 μm C18 Aurora column with Captive Spray emitter. Peptides were separated using a Bruker Nano-elute nUPLC at 400 nL/min with 0.1% formic acid (solvent A) and 100% acetonitrile, 0.1% formic acid (solvent B) and the following gradient conditions: 2% solvent B to 35% solvent B over 30 min. Runs were acquired in dia-PASEF mode (*Meier et al., 2020*) with an acquisition scheme consisting of four 25 m/z precursor windows per 100 ms TIMS scan. Sixteen TIMS scans, creating 64 total windows, layered the doubly and triply charged peptides on the m/z and ion mobility plane. Precursor windows began at 400 m/z and continued to 1200 m/z. The collision energy was ramped linearly as a function of ion mobility from 63 eV at 1/K0 = 1.5 Vs/cm$^2$ to 17 eV at 1/K0 = 0.55 Vs/cm$^2$.

Raw files acquired in DIA mode on the Fusion/Lumos instrument were analyzed with DIA-NN 1.7.12 (*Demichev et al., 2020*) using the following settings (protease: Trypsin/P, missed cleavages: 1, variable modifications: 1, peptide length range: 7–30, precursor m/z range: 300–1800, fragment ion m/z range: 200–1800, precursor false discovery rate [FDR]: 1). The N-term M excision, C carbamidomethylation, M oxidation, and RT profiling options were enabled and all other parameters were set to default. To generate a sample-specific spectral library, we initially used DIA-NN to create a large proteome-scale in silico deep learning-based library from the Uniprot human reference proteome (UP000005640, one protein per gene) with a list of common contaminants. This large spectral library was refined with deep sample specific chromatogram libraries. In short, equal amount of peptides from all U2OS cell lines (control, OCIAD1 knockdown, OCIAD2 knockdown, OCIAD1/2 double knockdown, and OCIAD1 knockdown rescued with wildtype OCIAD1) were pooled to create a master sample containing all peptides theoretically identifiable within our samples. To maximize the depth of our library, whole-cell lysate and mitochondrial pooled samples were processed separately. Deep chromatogram libraries were created from these pooled samples using six GPF DIA injections with a total of 52 overlapping 4 m/z-wide windows ranging from 400 to 1000 m/z as previously described (*Searle et al., 2018*). The resulting chromatogram libraries were used together with the

large predicted deep learning-based spectral library to generate a new highly optimized spectral library. This new spectral library was subsequently used to process our analytical samples.

Raw files acquired in diaPASEF mode on the timsTOF were analyzed similarly with DIA-NN (version 1.7.13 beta 1) using the following settings (protease: Trypsin/P, missed cleavages: 1, variable modifications: 1, peptide length range: 7–30, precursor m/z range: 300–1800, fragment ion m/z range: 200–1800, precursor FDR: 1, MS1 mass accuracy: 10 ppm, MS2 mass accuracy: 10 ppm). The N-term M excision, C carbamidomethylation, and M oxidation options were enabled and all other parameters were set to default. In short, we generated a deep learning-based predicted library from the Uniprot human reference proteome (UP000005640, one protein per gene) supplemented with N-terminal truncated CYC1 isoforms and a list of common contaminants. This large library was used to process all raw files from the gel slices analytical runs and generate a second and more optimized spectral library that includes ion mobility data. This new highly optimized spectral library was finally used to re-analyze all raw files.

DIA-NN output files were imported and analyzed in R using MaxLFQ values quantified from proteotypic peptides only (*Cox et al., 2014*). For whole-cell proteomics and immunoprecipitation analysis, only proteins identified in all the replicates of at least one sample were selected. Missing values were imputed using the 'MinDet' deterministic minimal value approach from the MSnbase package prior statistical analysis as described below.

## Confocal fluorescence microscopy

U2OS cells were grown on 12 mm round glass coverslips (#1.5) and stained for 30 min with 100 nM of Mitotracker DeepRed (Invitrogen, Carlsbad, CA, cat# M22426), washed with PBS, and fixed in 4% PFA in PBS for 20 min at room temperature. Cells were washed again with PBS, permeabilized for 10 min with 0.1% Triton X-100 in PBS, blocked with 5% bovine serum albumine (BSA) in PBS for 1 hr at room temperature, and immunolabeled with rabbit anti-OCIAD1 (Invitrogen, cat# PA5-20834, 1:10000) or rabbit anti-OCIAD2 (Invitrogen, cat# PA5-59375, 1:5000) antibodies for 1 hr at room temperature in 1% BSA in PBS. Cells were washed again in PBS and incubated with donkey anti-rabbit IgG conjugated with AlexaFluor 488 (Invitrogen, Carlsbad, CA, cat# A21206, 1:1000) in 1% BSA in PBS for 1 hr at room temperature. Finally, cells were washed again in PBS and mounted on glass slides with ProLong Glass antifade mounting medium (Invitrogen, Carlsbad, CA, cat# P36980). Images were collected using the spinning disk module of a Marianas SDC Real Time 3D Confocoal-TIRF microscope (Intelligent Imaging Innovations; Denver, CO) fitted with a 100×, 1.46 NA objective and a Hamamatsu (Japan) Orca Flash 4.0 sCMOS camera. Images were captured with SlideBook (Intelligent Imaging Innovations) and linear adjustments were made using ImageJ.

## Multiple sequence alignment

Multiple sequence alignment analysis was performed with the R package 'msa' (version 1.22.0) using the Clustal Omega method with default parameters.

## Statistical analysis

GO enrichment analysis was performed using the topGO R package (version 2.42.0) using the 'elim' method and Fisher's exact test (*Alexa et al., 2006*; *Grossmann et al., 2007*). Western blot densitometry results were analyzed using one-way analysis of variance followed by pairwise t-test with Benjamini and Hochberg (FDR) correction. For LC-MS/MS immunoprecipitation and whole-cell proteomics data, relative changes between conditions were analyzed using limma's function lmFit (*Ritchie et al., 2015*), followed by eBayes with FDR correction (*Phipson et al., 2016*). For whole-cell proteomics data, hierarchical clustering was performed using Euclidean distances of significant hit proteins. Error bars represent standard error and *$p<0.05$, **$p<0.01$, and ***$p<0.001$. All data were analyzed in R (version 4.0.3).

## Acknowledgements

We would like to thank Vadim Demichev for helping with DIA-NN analysis. We also want to thank James A Letts and María Maldonado for their valuable feedback on this manuscript. JN was supported by funding from the NIGMS (R37GM097432 and R01GM126081). JSW was supported by funding from HHMI and 1RM1HG009490. MJ was supported by funding from the NIH (grant

K99GM130964). JRF was supported by funding from the NIH (K99HL133372). FACS sorting at UC Davis is supported by the NIH (S100D018223). Min Y Cho coordinated resource sharing between labs and assisted with screen sample processing.

---

## Additional information

### Competing interests

Marco Jost: MJ consults for Maze Therapeutics. Jonathan S Weissman: JSW consults for and holds equity in KSQ Therapeutics, Maze Therapeutics, and Tenaya Therapeutics. JSW is a venture partner at 5AM Ventures and a member of the Amgen Scientific Advisory Board. The other authors declare that no competing interests exist.

### Funding

| Funder | Grant reference number | Author |
|---|---|---|
| National Institute of General Medical Sciences | R37GM097432 | Jodi Nunnari |
| National Institute of General Medical Sciences | R01GM126081 | Jodi Nunnari |
| National Heart, Lung, and Blood Institute | R00HL133372 | Jonathan Friedman |
| Howard Hughes Medical Institute | HHMI | Jonathan S Weissman |
| National Institute of General Medical Sciences | K99GM130964 | Marco Jost |
| National Human Genome Research Institute | 1RM1HG009490 | Jonathan S Weissman |
| National Institutes of Health | K99HL133372 | Jonathan Friedman |
| National Institutes of Health | S100D018223 | Jonathan Friedman |

The funders had no role in study design, data collection and interpretation, or the decision to submit the work for publication.

### Author contributions

Maxence Le Vasseur, Conceptualization, Data curation, Formal analysis, Validation, Investigation, Visualization, Methodology, Writing - original draft, Writing - review and editing; Jonathan Friedman, Conceptualization, Data curation, Formal analysis, Validation, Investigation, Visualization, Methodology, Writing - review and editing; Marco Jost, Resources, Software, Formal analysis, Visualization, Methodology, Writing - review and editing; Jiawei Xu, Formal analysis, Validation; Justin Yamada, Methodology, Writing - review and editing; Martin Kampmann, Resources, Software, Formal analysis, Methodology, Writing - review and editing; Max A Horlbeck, Resources, Software, Formal analysis; Michelle R Salemi, Formal analysis, Methodology; Brett S Phinney, Resources, Formal analysis, Methodology; Jonathan S Weissman, Resources, Data curation, Software, Formal analysis, Funding acquisition, Methodology, Writing - review and editing; Jodi Nunnari, Conceptualization, Supervision, Funding acquisition, Methodology, Writing - original draft, Project administration, Writing - review and editing

### Author ORCIDs

Maxence Le Vasseur (iD) https://orcid.org/0000-0001-6731-4379
Jonathan Friedman (iD) http://orcid.org/0000-0001-8155-2429
Marco Jost (iD) http://orcid.org/0000-0002-1369-4908
Jiawei Xu (iD) http://orcid.org/0000-0002-8534-2493
Martin Kampmann (iD) http://orcid.org/0000-0002-3819-7019

Max A Horlbeck http://orcid.org/0000-0002-3875-871X
Jodi Nunnari https://orcid.org/0000-0002-2249-8730

**Decision letter and Author response**
Decision letter https://doi.org/10.7554/eLife.67624.sa1
Author response https://doi.org/10.7554/eLife.67624.sa2

## Additional files

**Supplementary files**
• Supplementary file 1. List of sgRNA guides used in this study. Read counts and phenotypes for individual sgRNAs as well as gene-level phenotypes.

• Supplementary file 2. Results from the HHpred analysis.

• Supplementary file 3. List of primers and cloning strategy used in this study.

• Transparent reporting form

### Data availability

Mass Spectrometry files have been deposited to PRIDE under identifier numbers PXD025576, PXD025573, and PXD025682. Other data generated or analyzed during this study are included in the manuscript and supporting files.

The following datasets were generated:

| Author(s) | Year | Dataset title | Dataset URL | Database and Identifier |
|---|---|---|---|---|
| Le Vasseur M, Phinney BS, Salemi MR, Nunnari J | 2021 | Characterizing the role of OCIAD1 in the proteolytic processing of holocytochrome c1 and CIII2 assembly | http://www.ebi.ac.uk/pride/archive/projects/PXD025576 | PRIDE, PXD025576 |
| Le Vasseur M, Phinney BS, Salemi MR, Nunnari J | 2021 | Mapping perturbations in the cellular proteome of OCIAD1 and OCIAD2 knockdown U2OS cells | http://www.ebi.ac.uk/pride/archive/projects/PXD025573 | PRIDE, PXD025573 |
| Le Vasseur M, Phinney BS, Salemi MR, Nunnari J | 2021 | Mapping the interactome of OCIAD1 in human K562 cells | http://www.ebi.ac.uk/pride/archive/projects/PXD025682 | PRIDE, PXD025682 |

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
