## [Decision Letter]

**Acceptance summary:**

This study identifies a new assembly factor of the mitochondrial respiratory chain and provides an interesting link to the prohibitin scaffold complex of the mitochondrial inner membrane. The authors adopt a genome-wide human CRISPRi screen for selective inhibition of mitochondrial respiratory complex III and identify OCIAD1 as an important gene for mitochondrial function. The authors undertake proteomics and functional analysis leading them to suggest that OCIAD1 may function in a protease machine for the biogenesis of a complex III subunit. This study provides important new insights into the biogenesis of the mitochondrial respiratory chain in mammalian cells.

**Decision letter after peer review:**

Thank you for submitting your article "Genome-wide CRISPRi identifies OCIAD1 as a prohibitin client and regulator of mitochondrial Complex III assembly" for consideration by *eLife*. Your article has been reviewed by 3 peer reviewers, and the evaluation has been overseen by Klaus Pfanner as the Reviewing Editor and Vivek Malhotra as the Senior Editor. The reviewers have opted to remain anonymous.

Essential revisions:

1. The proteomics approaches are particularly powerful for identifying and supporting the findings. However, it would be much more useful for the spreadsheets reporting the MS data to be presented in a form that correlates with the figures. This is particularly so in volcano plots where the authors have only labelled those genes that are relevant to their study/hypotheses while the spreadsheets provide a list in alphabetical order of the gene names. Please revise.

2. The implication of the prohibitin complex in OCIAD1-facilitated processing of CYC1 by IMMP2L relies on circumstantial evidence. The authors show that OCIAD1 as well as IMMP2L interact with prohibitins, but not that the prohibitin complex plays a significant role in this pathway. To establish the relevance of the prohibitin interactions more robustly, the authors should provide evidence that knockdown or other interference with the prohibitin complex has an impact on CYC1 processing or they should soften their conclusion on this point.

*Reviewer #1:*

The authors undertook a CRISPRi genetic screen selecting cell lines where gene expression loss is affected by antimycin treatment under galactose growth conditions. In this way the screen selects against total respiration defects which would be able to persist on glucose media.

One uncharacterised gene – OCIAD1 – was highlighted for additional study due to the compromised growth of cells following its loss. The authors establish that OCIAD1 is a mitochondrial IMS-facing protein involved in the biogenesis of complex III of the respiratory chain.

They nicely identify that OCIAD1 assembles with the Prohibitin protease complex and also interacts with a number of proteins including the complex III subunit CYC1.

Loss of OCIAD1 impairs CYC1 stability and/or biogenesis with reduced protein present and a block in the processing of the MTS by IMMPL2. Since it was previously found that CYC1 processing is not critical for complex III function, the authors suggest that OCIAD1 may function also as a chaperone to stabilize newly imported CYC1 and presentation to the prohibitin-IMMPL2 complex.

I understand that this is a lot of work and the authors have uncovered some valuable insights. Some of the following should be considered to strengthen the findings and/or accessibility:

1. Is the role of OCIAD1 limited to CYC1? Given that GPD2 is also presented to IMMP2L the authors could establish whether GPD2 processing is also affected.

2. The blue native PAGE results are a little unclear following OCIAD1 depletion – particularly for complex I which seems to accumulate as a larger complex (Figure 2B). What is this and is it reproducible? Have the authors assessed respiration rates in the cell lines to characterise the defect more clearly?

3. The authors have performed proteomic analysis of OCIAD1 but choose to show a heat map with select genes. Can they also show a volcano plot to more clearly highlight the changes in the proteome following OCIAD1 loss?

4. The pull down of OCIAD1 in rescue vs knockdown lines in Figure 4B may lead to slightly misleading results as the comparison is between a pull down between a rescue line and an OCIAD2 depleted line. This means that the results can bias towards proteins that are more abundant in the rescue lines – so this will include complex III proteins. A more appropriate control in this case would be to compare the OCIAD1 pulldown against a pulldown with no-OCIAD1 antibodies. Also it seems that the level of OCIAD1 rescue may be greater that endogenous (Figure 4B) so the pull down could be done using control rather than rescue cells.

5. The reduction in complex III is not so high. I would suggest that if the authors made a clear knockout they may see a much stronger defect in complex III assembly since even small amounts of an assembly factor can compensate. If the authors grow the cells on glucose and do the knock down/out, they may also see a stronger CIII loss since the dependency of this enzyme is lost.

*Reviewer #2:*

The authors identified OCIAD1 using an unbiased genome-wide CRISPR screen for selective inhibition of CIII (antimycin sensitive). Having identified OCIAD1 as a major hit in this screen they performed a detailed biochemical analysis on the localization of the protein, its topology, its interactome within the CIII complex and the functional consequences of its genetic ablation. The work was performed in a very detailed manner and utilizing several approaches. The analysis of the localization of the protein and its topology is very solid and clarifies previously conflicting data on whether it is an OM or IM protein. The topology analysis using split GFP assays is convincing and will serve as a basis for further analysis of the functional role of the protein as a modulator of the cytc1 proteolysis. The finding that OCIAD1 is an assembly factor for CIII is important because in contrast to CI and CIV where several assembly factors have been characterized, this analysis for CIII is still lacking. The data comprise a very interesting body of work that will be important for future studies on potential other interactors of OCIAD1 and the interplay of this protein with other assembly factors for CIII.

I think the study is overall of high quality and the data very interesting.

*Reviewer #3:*

In their manuscript on OCIAD1, Le Vasseur et al. describe this protein as a novel regulator of respiratory complex III biogenesis in human cells. The authors identified OCIAD1 in a CRISPRi screen for genes that protect from or sensitize against complex III inhibition, and demonstrate that OCIAD1 is an inner membrane protein with its functional domains in the intermembrane space that interacts with the prohibitin complex and is required for the biogenesis of complex III. In the absence of OCIAD1 or in the point mutant F102A, the catalytic complex III subunit cytochrome c1 is not processed to its mature form, resulting in decreased levels and incorporation of this aberrant protein into the complex. The bipartite signal sequence of CYC1 requires processing by IMMP2L, resulting in removal of an N-terminal transmembrane segment. IMMP2L was previously identified as a prohibitin interactor, and the authors observed this peptidase also among the OCIAD1 interactome using a crosslinking approach. Since OCIAD1, but not the point mutant, interacts with CYC1, the authors conclude that OCIAD1 recruits the CYC1 precursor to the prohibitin complex and IMMT2L for processing, though the functional relevance of the prohibitin complex interactions has not been fully unravelled. Additionally, the study delineates that OCIAD2, despite its similarity to and interaction with OCIAD1, does not play a role in complex III biogenesis.

This exciting study employs a variety of elegant methods to identify and characterize the function of a novel assembly factor for human complex III. The data are of high quality , well presented, and they support the main conclusions.

1. The authors mention that OCIAD1 has homology to yeast Cox20. They should include the data for this conclusion and discuss the homology (including OCIAD2) especially in light of the report that yeast Cox20 is involved in Imp2-mediated processing of Cox2. This could lead to more widely relevant insights into determinants of IMP interaction and facilitation of processing versus CYC1 binding and a potential chaperone activity of OCIAD1.

2. As a control, the authors should demonstrate using another substrate that IMMP2L catalytic activity is not impaired in the absence of OCIAD1.

---

## [Author Response]

Essential revisions:1. The proteomics approaches are particularly powerful for identifying and supporting the findings. However, it would be much more useful for the spreadsheets reporting the MS data to be presented in a form that correlates with the figures. This is particularly so in volcano plots where the authors have only labelled those genes that are relevant to their study/hypotheses while the spreadsheets provide a list in alphabetical order of the gene names. Please revise.

We have modified the mass spectrometry spreadsheets to better connect them with the figures. Specifically, the revised spreadsheets are sorted in decreasing order of fold change as opposed to alphabetically. We also added a color scale to the fold change values to increase readability as well as a new column (-log_10_(adj.P.value)) to better reflect the values on the volcano plot y axis and indicated significant values using colored cells.

2. The implication of the prohibitin complex in OCIAD1-facilitated processing of CYC1 by IMMP2L relies on circumstantial evidence. The authors show that OCIAD1 as well as IMMP2L interact with prohibitins, but not that the prohibitin complex plays a significant role in this pathway. To establish the relevance of the prohibitin interactions more robustly, the authors should provide evidence that knockdown or other interference with the prohibitin complex has an impact on CYC1 processing or they should soften their conclusion on this point.

We softened our conclusion regarding the role of the prohibitin complex in the processing of CYC1 by adding the following sentence at the end of the discussion:

“However, further studies are needed to elucidate the functional significance of the OCIAD1 and prohibitin interaction in the maturation of CYC1.”

Reviewer #1:[…] I understand that this is a lot of work and the authors have uncovered some valuable insights. Some of the following should be considered to strengthen the findings and/or accessibility:1. Is the role of OCIAD1 limited to CYC1? Given that GPD2 is also presented to IMMP2L the authors could establish whether GPD2 processing is also affected.

We thank the reviewer for proposing this experiment that was also suggested by reviewer #3. As indicated in Figure 5—figure supplement 1, we did not observe processivity defects in GPD2 or AIF, two other putative IMMP2L substrates. These results suggest that OCIAD1 specifically regulates CYC1 processing but not the processing of other IMMP2L substrates. We added these results in Figure 5—figure supplement 1 along with the following sentence:

“The processing of GPD2 and AIF, two additional substrates of IMMP2L (Lu et al., 2008; Yuan et al., 2018), were unaffected in OCIAD1 knockdown cells as assessed by immunoblotting, suggesting that the maturation defect is specific to CYC1 (Figure 5—figure supplement 1D, E).”

We added the GPD2 and AIF antibodies used in the method section.

2. The blue native PAGE results are a little unclear following OCIAD1 depletion – particularly for complex I which seems to accumulate as a larger complex (Figure 2B). What is this and is it reproducible? Have the authors assessed respiration rates in the cell lines to characterise the defect more clearly?

This is good observation. The different bands likely represent various higher order configurations of supercomplex assemblies. While Figure 2B suggests that CI assembles into larger supercomplex entities in OCIAD1 knockdown cells, this effect shows high variability within and between samples, and trends towards being more pronounced in K562 than U2OS cells. This is better shown in Author response image 1 which displays the uncropped Figure 2B panel (left) as well as BN-PAGE results showing the organization of CI-containing supercomplexes in U2OS cells (right; not shown in the manuscript and contains one sample unrelated to the current project). These results are consistent with our conclusion that OCIAD1 is not a major player in the organization of higher order supercomplex assemblies. We decided against including these images in the main manuscript as we felt that commenting on the slight variation shown in Figure 2B would distract from the main conclusion that OCIAD1 significantly affects CIII assembly.

We have not systematically assessed respiratory rates in these cell lines. While interesting, we do not think that monitoring cellular respiration will provide additional insight on the function and/or mechanism of action of OCIAD1 in the context of CYC1 processing.

3. The authors have performed proteomic analysis of OCIAD1 but choose to show a heat map with select genes. Can they also show a volcano plot to more clearly highlight the changes in the proteome following OCIAD1 loss?

The heat map displayed in Figure 4—figure supplement 4 is unbiased and contains all the proteins that were shown to be differentially regulated in at least one of the cell line and is therefore not limited to hand-picked genes. While volcano plots are visually very informative, they are limited to the comparison of two conditions. In this instance, we felt that a heat map capturing the relationship between all 5 different cell lines better than a combination of volcano plots comparing each cell line to each other.

4. The pull down of OCIAD1 in rescue vs knockdown lines in Figure 4B may lead to slightly misleading results as the comparison is between a pull down between a rescue line and an OCIAD2 depleted line. This means that the results can bias towards proteins that are more abundant in the rescue lines – so this will include complex III proteins. A more appropriate control in this case would be to compare the OCIAD1 pulldown against a pulldown with no-OCIAD1 antibodies. Also it seems that the level of OCIAD1 rescue may be greater that endogenous (Figure 4B) so the pull down could be done using control rather than rescue cells.

We would like to refer the reviewer to our untargeted whole proteome quantification shown in Figure 4—figure supplement 4 and emphasize that the number of proteins significantly altered by rescuing OCIAD1 expression is almost negligeable. Therefore, it is unlikely that our immunoprecipitation data were skewed towards proteins more abundantly expressed in the rescue cell line versus control. While we acknowledge that OCIAD1 levels are slightly higher in the rescue cell line (141.20 ± 6.07% of control as shown in Figure 2A), a majority of rescued OCIAD1 is associated with the prohibitin complex (as shown in Figure 4B) and is therefore likely fully integrated into its native molecular environment. In addition, our OCIAD1 interactome largely overlaps with prohibitin interactomes published by other groups, further supporting the idea that OCIAD1 is behaving similar to endogenous OCIAD1 in the rescued cell line. Immunoprecipitating OCIAD1 in the rescue cell line allowed us to use the same samples in our comparison against the OCIAD1 knockdown cells (shown in Figure 4A) and the F102A mutant (shown in Figure 5C) as the mutant was also similarly generated by OCIAD1 expression rescue in OCIAD1 knockdown cells.

5. The reduction in complex III is not so high. I would suggest that if the authors made a clear knockout they may see a much stronger defect in complex III assembly since even small amounts of an assembly factor can compensate. If the authors grow the cells on glucose and do the knock down/out, they may also see a stronger CIII loss since the dependency of this enzyme is lost.

We agree with the reviewer and acknowledge that the deficit in CIII assembly might be stronger in a complete OCIAD1 knockout system. However, we do not anticipate a total OCIAD1 knockout to entirely abrogate CIII assembly. As we demonstrated in the manuscript, the processing of CYC1 is dispensable for its incorporation into CIII assemblies and CIII assemblies containing immature CYC1 are also likely to form in OCIAD1 KO cells.

Reviewer #3:[…] 1. The authors mention that OCIAD1 has homology to yeast Cox20. They should include the data for this conclusion and discuss the homology (including OCIAD2) especially in light of the report that yeast Cox20 is involved in Imp2-mediated processing of Cox2. This could lead to more widely relevant insights into determinants of IMP interaction and facilitation of processing versus CYC1 binding and a potential chaperone activity of OCIAD1.

This is a great suggestion. We included the results of the analysis in a supplementary file and modified the discussion regarding the HHpred analysis as follows:

“OCIAD1 and its related paralog OCIAD2 are highly conserved in metazoans. […] The high homology between COX20 and OCIAD1, but not OCIAD2, is consistent with the functional specialization of both OCIAD paralogs and indicates that OCIAD1 serves a conserved function by facilitating the proteolytic processing of CYC1.”

2. As a control, the authors should demonstrate using another substrate that IMMP2L catalytic activity is not impaired in the absence of OCIAD1.

This is a great suggestion and we would like to refer the reviewer to our answer to reviewer #1 above who had the same comment.